# FINDINGDORY: A BENCHMARK TO EVALUATE MEMORY IN EMBODIED AGENTS

## ABSTRACT

Vision-Language models (VLMs) have recently demonstrated impressive performance in planning and control tasks, driving interest in their application to robotics. Yet their deployment in embodied settings remains limited by the challenge of incorporating long-term experience, often spanning multiple days and represented by vast image collections. Current VLMs typically handle only a few hundred images at once, underscoring the need for more efficient mechanisms to manage long-term memory in embodied contexts. To meaningfully evaluate these models for long-horizon control, a benchmark must target scenarios where memory is essential. Existing long-video QA benchmarks neglect embodied challenges like object manipulation and navigation, which require low-level skills and fine-grained reasoning over past interactions. Moreover, effective memory integration in embodied agents involves both recalling relevant historical information and executing actions based on that information, making it essential to study these aspects together. In this work, we introduce FINDINGDORY, a new benchmark for long-range embodied tasks in the Habitat simulator. FINDINGDORY evaluates memory-centric capabilities across 60 tasks requiring sustained engagement and contextual awareness in an environment. The tasks can also be procedurally extended to longer and more challenging versions, enabling scalable evaluation of memory and reasoning. We further present baselines that integrate state-of-the-art closed-source and fine-tuned open-source VLMs with low-level navigation policies, assessing their performance on these memory-intensive tasks and highlighting key areas for improvement.[1]

## 1 INTRODUCTION

Memory is a fundamental capability for intelligent agents that allows them to recall past experiences, adapt to dynamic environments, and make informed decisions over extended timescales. In humans and animals, it plays a crucial role in navigation, reasoning, and goal-directed behavior. As we strive to develop more capable embodied systems, equipping them with robust memory mechanisms is essential, particularly in settings where agents must process high-dimensional multimodal inputs to interact with their surroundings. The ability to store and retrieve relevant information is the key to unlocking sophisticated behaviors, from household assistance to autonomous exploration.

Recent advances in Vision-Language Models (VLMs) have significantly improved high-level reasoning and planning for embodied tasks. These models leverage large-scale multimodal training to exhibit strong scene understanding and action-guided perception. However, most existing VLM applications focus on short-term or static tasks, such as image captioning or Visual Question Answering (VQA), which do not require sustained memory usage. Extending these models to long-horizon embodied control introduces new challenges: embodied agents must integrate observations over time, recall past experiences, and act accordingly within their environment.

While recent efforts have sought to extend long-context understanding in VLMs, primarily through video-based QA or long-document comprehension (Song et al., 2023; Xiong et al., 2025; Ma et al., 2024), these approaches fail to fully capture the complexities of long-term memory in decision-making. Many existing QA benchmarks either rely on multiple-choice formats, which enable guessing, or require extensive human annotation to curate question-answer pairs that require models to

---

[1]Website: `https://findingdorybenchmark.github.io/`

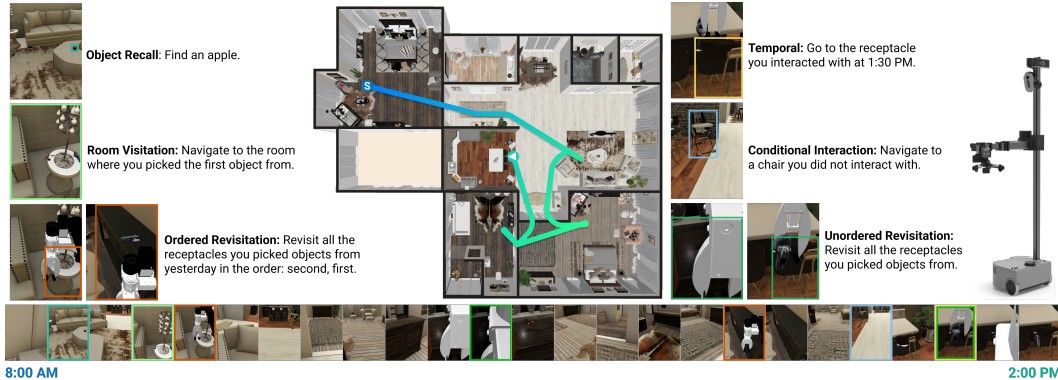

Figure 1: FINDINGDORY: We propose a benchmark for evaluating agents' memory by constructing scenarios that require reasoning over previously collected experiences. In these scenarios, an agent receives logs of prior interactions in indoor environments and must complete navigation, pick, and place instructions. The figure shows a sample episode: first, a privileged agent with full environment access executes randomly sampled navigation and manipulation tasks. Then, the memory-evaluating agent is placed in the same environment and tasked with completing instructions that rely on the privileged agent's logs to decide where to navigate or which object to retrieve.

attend to long video sequences. In contrast, we propose embodied memory tasks—such as *"navigating to a soft toy that you did not rearrange yesterday"* or *"find the second object picked up earlier"* that demand fine-grained recall and multi-hop reasoning over past observations. These tasks provide a more rigorous test of memory capabilities, requiring agents to track and respond to environmental changes rather than passively perceive static information.

In this paper, we introduce FINDINGDORY, a benchmark designed to evaluate long-horizon memory in highly photorealistic simulated environments. Our benchmark consists of **60 diverse navigation tasks**, which we categorize along various axes of memory requirements, that require both long-range temporal and spatial reasoning while discouraging heuristic shortcuts or random guessing. To ensure that performance is directly tied to memory utilization rather than simple perception, we curate scenarios where agents must recall past interactions to succeed. Additionally, our benchmark features dynamic environments - where agents modify the scene - making it essential to reason over changing contexts. Furthermore, FINDINGDORY is procedurally extensible, allowing tasks to scale in complexity as VLMs improve, ensuring continued relevance as embodied AI progresses.

Unlike standard QA datasets that rely on subjective human annotations, our benchmark leverages photorealistic simulation to enable automatic evaluation of memory-based navigation. These tasks go beyond static recall since they require agents to perform complex spatio-temporal reasoning over past interactions. For instance, tasks such as *"navigate to any object that you interacted with yesterday"* demand not only accurate memory retrieval, but also strategic decision-making to identify and navigate to the *nearest* valid target from where the agent is currently positioned. In FINDINGDORY, we introduce metrics that quantify task completion efficiency and not just successful recall.

Our key contributions are:

1. A benchmark for evaluating long-horizon memory in embodied decision-making, featuring 60 diverse navigation tasks in highly realistic indoor environments.
2. A comprehensive evaluation of VLM based high-level policies combined with low-level navigation policies, analyzing their performance and limitations in memory-intensive tasks.
3. A systematic and extensible evaluation framework with metrics that enable future advancements in memory-efficient embodied agents.

## 2 RELATED WORK

In this section, we review related work on long-video question answering and embodied AI benchmarks that incorporate historical experience in their problem formulation. We defer the discussion of different approaches for augmenting policies with memory-like mechanisms to appendix C.1.

Table 1: Comparison of FINDINGDORY with popular memory-focused embodied AI benchmarks.

| Benchmark | Photorealism | Semantic Reasoning | Task Specification | Template Categories | # Resulting Tasks | Memory Length ($\pi^*$) | Isolates Memory |
|---|---|---|---|---|---|---|---|
| MemoryMaze (Pašukonis et al., 2023) | ✗ | ✗ | One-hot | ✗ | 1 | 500-1000 steps | ✗ |
| MemoryGym (Pleines et al., 2023) | ✗ | ✗ | One-hot | ✗ | 3 | 128-512 steps (Extensible) | ✗ |
| MultiON (Wani et al., 2020) | ✓ | ✗ | One-hot | ✗ | 1 | 500 steps | ✗ |
| SIF (Min et al., 2024) | ✓ | ✓ | Language | 3 | 240 (val/test) | ✗ (Uses semantic map) | ✗ |
| OpenEQA (Active) (Majumdar et al., 2024) | ✓ | ✓ | Language | ✗ | 1600 | 75 steps | ✗ |
| GoatBench (Khanna* et al., 2024) | ✓ | ✓ | Language + Image | ✗ | 1800 − 3600 (val) | 500 steps (Static Env) | ✗ |
| Excalibur (Zhu et al., 2023) | ✓ | ✓ | Language | 11 | 21 | Requires exploration | ✗ |
| Ours | ✓ | ✓ | Language | 11 | ≈ 6000 (val) | 500-3500 steps (Extensible) | ✓ |

## 2.1 VIDEO QA BENCHMARKS

Recent analyses of video QA benchmarks (Mangalam et al., 2023; Buch et al., 2022) show that many tasks can be solved by attending to key frames, not full video sequences, suggesting that current benchmarks may favor keyframe selection over evaluating long-term memory.

Long-context evaluations for foundation models also focus on "needle-in-a-haystack" tasks, evaluating retrieval of specific details rather than integrating information across many segments (Zhang et al., 2024). While recent video benchmarks feature more challenging questions (Mangalam et al., 2023; Li et al., 2023; Wang et al., 2024; Fu et al., 2024; Song et al., 2023; 2024; Li et al., 2023; Rawal et al., 2024; Yang et al., 2024; Zhou et al., 2024; Majumdar et al., 2024), they adopt multiple-choice QA formats which are not suitable for evaluating fine-grained decision-making over long histories. They may also allow models to succeed through guessing and are not extensible since they require substantial human annotation effort.

## 2.2 EMBODIED AI BENCHMARKS

Various benchmarks have been proposed to evaluate multi-modal language models in embodied settings where agents must leverage their experience in environments to improve future decision-making (Wani et al., 2020). Many focus on embodied question answering (EQA), such as HM-EQA (Ren et al., 2024), which requires active exploration, and Open-EQA (Majumdar et al., 2024), which tests VLMs in a passive setting with pre-collected episode histories.

Notably, Open-EQA found that a *blind* LLM baseline—one that ignores visual input—still performed well, suggesting that multiple-choice evaluation can introduce biases and reduce the need for genuine memory recall. Other benchmarks study the situated instruction following setting where agents must gather information to interpret ambiguous instructions (Sashank Dorbala et al., 2024; Ma et al., 2023; Min et al., 2024). However, many allow agents to sidestep memory challenges by precomputing relevant objects and locations based on recent history, effectively turning potentially memory-intensive tasks into structured instruction-following problems. Memory-Maze and Memory-Gym (Pašukonis et al., 2023; Pleines et al., 2023) focus on partially observable decision-making with history-based state modeling, but their simplistic visuals and small state spaces limit their relevance to complex real-world memory settings.

In table 1, we compare our proposed benchmark to the most related benchmarks within embodied AI and RL research with respect to the following desired properties:

1. **Photorealism**: Measures the visual complexity of environments, affecting the meaningful evaluation of foundation models trained on real-world data.
2. **Scope of reasoning needed over memory**: Distinguishes between benchmarks that require recalling diverse semantic information v.s. a narrow set of attributes (e.g. object locations).
3. **Task Specification**: Evaluates whether tasks are defined using flexible language-based descriptions, enabling richer memory challenges.
4. **Memory horizon length**: Adapting the definition from Ni et al. (2023), this measures the temporal scope over which memory must be maintained for optimal task performance. Unlike static benchmarks requiring localized frame recall, dynamic environments demand reasoning over events distributed across extended observation sequences. This metric becomes difficult to quantify when tasks require substantial exploration phases.
5. **Memory isolation**: Whether benchmark performance isolates memory evaluation from confounding factors, particularly exploration requirements, which represent orthogonal challenges to memory recall and reasoning.

Our benchmark isolates memory evaluation by avoiding confounding related to exploration and focuses on careful and extensible task curation with automated metrics, requiring agents to recall and reason over long-horizon environmental observations.

## 3 FINDINGDORY

In this section, we introduce our embodied memory benchmark, FINDINGDORY, built on top of the **Habitat simulator** Savva et al. (2019). Habitat offers photorealistic visuals, diverse household scenes, and object-rich environments, making it a practical and **transferable testbed** for evaluating memory-augmented embodied agents. Since many vision-language models (VLMs) are pretrained on real-world image distributions, they can directly interpret observations from Habitat, enabling realistic and scalable evaluation of long-term decision-making.

We first describe the overall structure of FINDINGDORY and the suite of tasks it comprises, then provide implementation details.

### 3.1 BENCHMARK DESIGN AND TASKS

The goal of FINDINGDORY is to evaluate an agent's ability to retain and utilize past experiences to efficiently complete future tasks. To achieve this, we design a two-phase setup: an *experience collection phase*, where scripted oracle agents autonomously gather experiences in the environment, and an *interaction phase*, where agents are evaluated on their ability to recall and act upon that history. This decomposition isolates memory from exploration, enabling standardized evaluation.

During **experience collection**, oracle agents pick up and place objects onto designated receptacles (e.g., a table, chair, or bed), introducing meaningful state changes into the environment. Episodes span 400–3500 frames with 2–11 pick-and-place interactions (see appendix B.2). Once collection ends, the **interaction phase** begins: the agent is given access to its recorded history (images, pose, past actions) and tasked with following instructions that require integrating long-term memory into decision-making.

Tasks are instantiated from a set of **templated instructions** designed to probe diverse memory challenges. These cover **spatial**, **temporal**, and **semantic** reasoning: e.g., recalling which object was moved, when it was moved, or how it was interacted with; identifying rooms visited or farthest locations; and reasoning over attributes such as shape, color, or material. We also introduce **time-usage memory** tasks (e.g., longest room exploration, object with longest manipulation). A smaller subset of **attribute-based tasks** involves referring to objects indirectly via observed properties (e.g., "go to the striped object you interacted with previously"), helping distinguish methods that can flexibly reconstruct fine-grained visual details from those relying on static summaries. In total, 60 templates are used to generate scene-specific task instances (see table 5).

**Single vs. Multiple Goals.** Most tasks involve single-goal instructions, but some require reaching multiple destinations in sequence. Even single-goal tasks can be challenging, demanding agents recall what was *not* interacted with or trace object movements across rearrangement states.

**Extensibility.** The task suite is procedurally extensible: adjusting the complexity of experience collection (e.g., number of interactions) scales the temporal reasoning space and memory dependencies. We further group tasks into interpretable categories for analysis (see table 2), making FINDINGDORY one of the most visually complex and extensible embodied memory benchmarks to date.

### 3.2 IMPLEMENTATION DETAILS

**Simulation Setup and Dataset Composition.** We build FINDINGDORY using the Habitat simulator (Savva et al., 2019; Szot et al., 2021) and scenes from the Habitat Synthetic Scenes Dataset (HSSD) (Khanna* et al., 2023). Our benchmark uses 107 scenes from the training split and 30 from the validation split, with 1,478 training and 100 validation episodes. We populate these scenes with the same object sets used in OVMM (Yenamandra et al., 2023b). Across these episodes, we include 839/247 object instances spanning 84/72 distinct categories in the `train/val` splits, respectively. Objects are rearranged during the experience collection phase to induce temporally grounded memory challenges. Additionally, agents interact with 17/16 distinct receptacle categories in the `train/val` splits (e.g., tables, chairs, beds). To support attribute-based tasks (see table 2), we generate semantic descriptions of object instances using GPT-4o. We prompt the model with multi-view images of each object to extract attributes such as shape, color, material, and functionality. These descriptions are used to populate the object attribute task templates (5 total), which are limited to the validation set and undergo manual quality review. In total, our benchmark includes 79,213 training and 5,876 validation task instances. More details in appendix B.

Table 2: Task categories with example prompts and challenges. See appendix H for more examples.

| | Task Category | Example Instructions | Memory Requirement |
|---|---|---|---|
| **Spatial** | Object Recall | Navigate to a {*category*}. | Object was previously seen or needs exploration. |
| | Interaction | Navigate to any object that you did not interact with yesterday. | Differentiate between interacted & uninteracted objects. |
| | Conditional Interaction | Navigate to a {*receptacle_category*} you picked an object from. | Object categories, object-receptacle relationships. |
| | Object Attribute | Navigate back to a {*target_color*} colored object that you interacted with yesterday. | Object attributes other than category. |
| | Spatial Relationship | Navigate to the object which you interacted with which is the farthest from your current location. | Spatial relationships to objects. |
| | Room Visitation | Navigate to a room that you did not visit yesterday. | Room boundaries and past interactions. |
| **Temporal** | Interaction Order | Navigate to the object you interacted with immediately after {*object_category*}. | Sequence of objects interactions. |
| | Time-Based | Navigate to the object that you interacted with at {*HH:MM*} yesterday. | Time of interaction & final position of objects. |
| | Duration Tracking | Navigate to the object which took the longest time to rearrange. | Duration of interactions. |
| **Multi-Goal** | Unordered Revisitation | Revisit all the receptacles you picked objects from yesterday. | Past locations of multiple objects and best visitation order. |
| | Ordered Revisitation | Revisit all the objects you interacted with yesterday in specific order. | Similar to unordered revisitation but with strict sequencing. |

**Agent Setup.** We adopt the Hello Robot Stretch embodiment (Kemp et al., 2022) in our simulation. The agent is equipped with RGB and depth sensors (resolution $640 \times 480$) and a GPS+Compass sensor. The agent can execute one of four discrete navigation actions: MOVE_FORWARD (0.25m), TURN_LEFT / TURN_RIGHT (by $10°$) and a STOP action. During the experience collection phase, the agent can also manipulate its arm joints (extension, lift, yaw, pitch, roll) and activate a manipulation_mode for pick-and-place actions, which rotates the base ($90°$) and directs the head camera toward the gripper.

**Procedural Generation and Verification using PDDL.** We operate over a large set of diverse task instructions (see table 2) by encoding them into templates using the PDDL specification system (Aeronautiques et al., 1998; Szot et al., 2024). Specifically, each template specifies *entities* (objects and receptacles) with *properties* such as interaction order, target color, and start/goal receptacle. These entities are combined using *predicates* that map to pythonic functions which can be evaluated against any simulator state. *Goal conditions* compose these predicates with *quantifiers* (e.g., EXISTS, FOR_ALL_UNORDERED) to express complex logical relationships between multiple predicates. The defined instructions templates are *procedurally* expanded by sampling valid entity bindings based on the entity properties which serve as constraints. The bindings are finally substituted into the goal conditions to generate the FINDINGDORY dataset and enable scalable, automated verification for each episode without hand-coding low-level rules.

## 4 EXPERIMENTS

We evaluate several baseline approaches on FINDINGDORY and assess their ability to complete memory-intensive tasks using a combination of vision-language reasoning and navigation. Below, we describe the baseline architecture, implementation details, and evaluation metrics.

### 4.1 EVALUATED APPROACHES

To establish strong baselines, we adopt a hierarchical architecture (fig. 15) comprising two modules: (1) a high-level goal selection module, which processes the interaction history to select a sequence of goal frames, and (2) a low-level navigation policy, which executes actions to reach the selected frame. This architecture enables reasoning over long-horizon multimodal experience while delegating fine-grained control to a learned policy. Inspired by prior work (Xu et al., 2024), we avoid

directly predicting actions using VLMs due to their poor performance in continuous control, and instead restrict them to high-level goal prediction.

### 4.1.1 HIGH-LEVEL GOAL SELECTION

**Video LM Agent.** This agent receives the full interaction video along with a task instruction. Each frame in the video is annotated with its index and the time of day at which it was captured. The agent is tasked with predicting the index of a frame from the video that corresponds to a viable goal state. Specifically, the predicted frame should represent a viewpoint from which, if the agent were to navigate back to the same camera position, it could successfully complete the task. Although the model outputs a single frame index, selecting the correct one often requires reasoning over multiple temporally scattered observations, including frames where the agent may not be actively interacting. Since different VLMs are capable of handling different numbers of frames in context, we subsample the frames passed to each model based on what empirically performs best. We evaluate both proprietary (Gemini-2.0-Flash (Team et al., 2024), GPT-4o) and open-source (Qwen2.5-VL (Bai et al., 2025), Gemma-3 (Team et al., 2025), GLM-4.1V-Thinking (Hong et al., 2025)) vision-language models capable of processing long video contexts. See appendices D.1 to D.3 for more details.

**Textual Memory Agent.** Similar to (Min et al., 2024; Anwar et al., 2024), we also consider an explicit, memory-building baseline which breaks down the video frames into chunks and uses a VLM to generate relevant textual summaries for each video chunk. After this guided "captioning" of the entire video sequence, an LLM selects goal frame indices by reasoning over the generated textual descriptions. See appendix D.4 for more details.

**Supervised Fine-Tuning (SFT).** We additionally explore a supervised fine-tuning approach where the high-level VLM is trained to predict goal frame indices, given the task prompt and interaction history. Since multiple frames may validly represent the goal (e.g., several consecutive frames showing an object being placed), we supervise the model on the full list of acceptable target frames. At test time, a single frame is sampled from the list of predictions and is considered correct if it matches any of these valid frames. This setup enables the model to learn task-specific cues from examples and serves as a stronger baseline than zero-shot prompting, particularly for tasks requiring subtle temporal grounding. We provide additional details in appendix D.5.

### 4.1.2 LOW-LEVEL NAVIGATION

To execute the high-level plan, we use a low-level policy that takes the selected goal frame and outputs discrete actions: `MOVE-FORWARD`, `TURN-RIGHT`, `TURN-LEFT`, and `STOP`. Our primary policy is the LSTM-based controller from OVRLv2 (Yadav et al., 2023), trained to reach image goals from egocentric RGB-D observations (see appendix E.1) (ImageNav). We also evaluate a mapping-based policy that deterministically navigates to the location corresponding to the goal frame using a global map, without relying on learned visual features (see appendix E.2).

**Solvability.** The action space of all our high-level baselines is constrained to frames observed during experience collection, which is insufficient for some tasks. For example, in the task *"Find an apple that you did not interact with yesterday"*, if the agent never closely approachs that apple during experience collection, then no frame in the history can serve as a goal frame. This reflects a limitation of the baseline's goal-parameterization rather than a limitation of the benchmark itself. To establish an upper bound on task difficulty for our current baselines, we quantify the percentage of episodes that could theoretically be solved through optimal frame selection from interaction history, assuming perfect navigation (teleportation) to the selected waypoint. This upper bound is the dashed line (`Oracle`) in fig. 2.

### 4.2 EVALUATION METRICS

We benchmark agent performance on the FINDINGDORY tasks using **Low Level Success Rate** (`LL-SR`), assessing whether the agent finds the right target entities, and **Low Level Success-weighted-by-Path-Length** (`LL-SPL`), evaluating if the agent selects and navigates to the optimal entity using the shortest possible path. Since we use a hierarchical baseline, we additionally introduce a **High-Level Policy Success Rate** (`HL-SR`), and **High-Level Policy SPL** (`HL-SPL`) which measures the accuracy and efficiency of the high-level policy in selecting the correct and closest goal frames for navigation. To further analyze failure modes, we also report two relaxed variants:

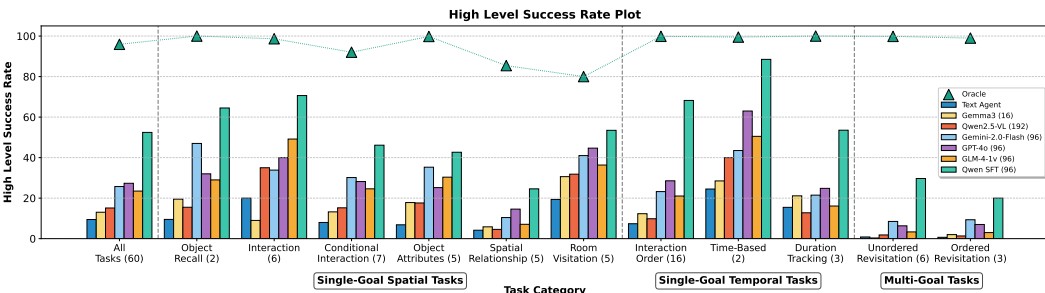

Figure 2: FINDINGDORY High-Level Success Rate Plot: We evaluate variety of baselines on the high-level task of FindingDory. The numbers in parentheses for each VLM denote the optimal frame budget used the model. Proprietary VLMs such as `Gemini-2.0-flash` and `GPT-4o` show little success across most FINDINGDORY task categories. Agents struggle especially on multi-goal tasks where multiple subgoals must be achieved sequentially from the interaction videos collected during Phase 1. The `Qwen-SFT` baseline (see appendix D.5), trained to predict ground-truth frame indices, performs best but still reaches only $\approx 50\%$ on high-level tasks.

**Distance-to-Goal-Only SR** (`DTG-SR`), which relaxes the requirement that the target object must be visible in the selected frame, and **Semantic Coverage SR** (`SC-SR`), which ignores spatial proximity. Additional metrics, success criteria, and threshold specifics are provided in appendix B.3.

## 5 RESULTS

In this section, we analyze the performance of the previously outlined baselines on FINDINGDORY tasks, presenting results grouped according to the task categories described in section 3.1.

**State-of-the-Art VLMs struggle on the FINDINGDORY benchmark.** We first evaluate different VLMs on the high-level task and report success rates in fig. 2, observing consistently low performance across all task subsets (see table 4 for numbers). GPT-4o achieves the highest success at 27.3%, followed by Gemini-2.0-Flash at 25.7%, and GLM-4.1V-Thinking at 23.5%. The non-reasoning open-source models Qwen2.5-VL and Gemma-3 score between 13% and 16%. The text-based agent performs the worst overall, underscoring that textual memory representations alone are insufficient for disambiguating visual goals or reasoning over complex visual histories. Notably, the supervised fine-tuned (SFT) variant outperforms all frozen VLMs, achieving an average improvement of 25% across tasks, with considerable improvement on temporal and multi-goal tasks.

In single-goal tasks, all methods struggle particularly with the spatial relationship subset, with GPT-4o achieving only 14.5% success. These tasks require reasoning about relative distances between entities and the agent, an ongoing challenge noted in prior work (Yang et al., 2024). In contrast, room identification tasks yield the highest performance, likely due to the small number of distinct rooms and the lenient success criteria of reaching any location within the correct room. Performance on conditional interaction and object attribute tasks is comparable, suggesting that current VLMs handle both category-based queries (e.g., "apple") and attribute-based descriptions (e.g., "red food") with similar effectiveness.

Temporal tasks pose additional challenges to the existing VLMs, with models consistently failing to identify the correct order of interactions, since it requires multi-hop reasoning. Duration tracking is similarly difficult as it requires inferring elapsed duration based on specific points in the video. In contrast, for the time-based tasks, we observe that baselines perform better, as these tasks only require single frame retrieval based on the timestamps in the question, without additional reasoning.

**Poor performance on multi-goal tasks.** From fig. 2, we observe that all VLM baselines achieve near-zero performance on most multi-goal tasks. This indicates that VLMs struggle to track and recall multiple object–receptacle interactions. In our multi-goal evaluation, we employ a single-shot prompt, requiring the high-level VLM agent to predict all target frame indices in one response, primarily to avoid repeated inferences over long sequences given the large number of frames. Qualitatively, we find that VLMs often fail to predict even the correct number of targets needed to solve a task. While our trained Qwen model performs somewhat better, reaching around 20% accuracy, it still falls far short of the attainable 99% success rate on this task.

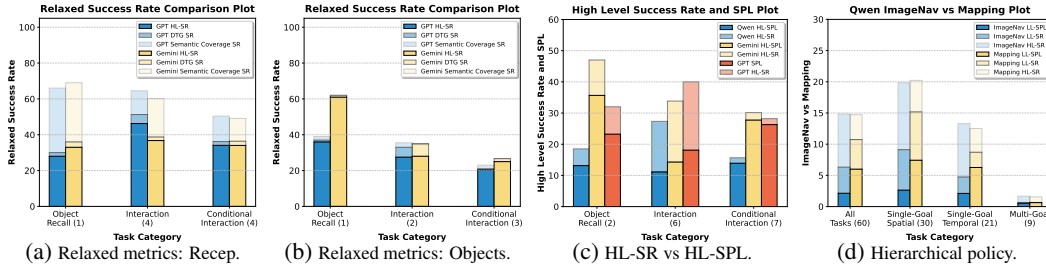

(a) Relaxed metrics: Recep.    (b) Relaxed metrics: Objects.    (c) HL-SR vs HL-SPL.    (d) Hierarchical policy.

Figure 3: **(a, b) Relaxed metrics for object and receptacle tasks.** Models show substantial gains on receptacle-based tasks under relaxed metrics, whereas tasks targeting small objects exhibit little improvement. **(c) Success Rate vs. SPL.** For tasks where the VLM may select among multiple correct frames, all models sometimes choose suboptimal ones, reducing the `HL-SPL` metric. **(d) Hierarchical policy on FINDINGDORY.** When Qwen is paired with an ImageNav or Mapping agent, overall SR and SPL drop markedly (as reflected in `LL-SR/LL-SPL`). See fig. 18 for decoupled visualizations.

**VLMs can identify target receptacles but not precisely localize them.** We analyze VLM performance on tasks where the valid entities are receptacles versus where they are objects. As shown in fig. 3a, the `Semantic Coverage-SR (SC-SR)` metric (see appendix B.3) is substantially higher for receptacle-based tasks. This indicates that VLMs often select frames where the receptacle is visible, but the agent remains far from it. In contrast, distance-threshold success (`DTG-SR`) improves negligibly, suggesting that very few cases exist where the agent is close to a receptacle but not facing it. For smaller objects, no such improvement is observed (see fig. 3b), since small objects are rarely visible from longer distances. Taken together, these results highlight a key limitation of current VLMs: they can semantically detect large targets but fail to construct spatially grounded representations necessary for precise localization.

**VLMs struggle with identifying the closest entity that solves the task.** In Fig. 3c we report the `HL-SPL` metric (section 4.2) which captures the efficiency with which the high level VLM-based modules select goal frames. During the experience collection phase, the agent can interact with multiple object instances of the same category due to which a task such as *"Navigate to a squeezer you interacted with"* could have multiple valid entities that the agent can navigate to. We observe that for such tasks there is a wide gap between `HL-SR` and `HL-SPL` (upto $50\%$ in case of interaction-based tasks) for all model variants. This suggests that although the VLMs can correctly recognize a valid entity to solve the task, they still fail to perform a fine-grained spatial analysis of the multiple valid entities being rearranged dynamically in the video. Thus, they cannot spatially locate the valid entity nearest to its current position.

**Policy decomposition causes performance degradation.** To evaluate the hierarchical agent introduced in section 4.1, we use `Qwen2.5-VL-7B` as the high-level VLM module and pair it with either an ImageNav policy or a mapping-based policy as the low-level agent. The low-level policy is triggered only when the high-level VLM correctly identifies a subgoal. As shown in fig. 3d, `LL-SR` drops sharply for the ImageNav policy. The same policy achieves 78% success on its native ImageNav validation episodes (see Appendix E.1), confirming it performs well when goal images match its training distribution. The degradation arises from distribution shift: VLM-selected goals often come from pick–place interactions with atypical viewpoints and limited visual cues, unlike the goals used during ImageNav training. By contrast, the mapping-based navigation policy is more robust, suffering only a 25% relative drop in SR. However, the SPL of both methods remains in the single digits, underscoring the importance of FINDINGDORY in exposing these shortcomings and motivating the development of more reliable memory and navigation strategies.

**Low level navigation policy failures.** From Figure 4b, we observe that the low-level image-goal navigation policy can fail due to invoking the termination (stop) action incorrectly or a navigation routine timeout (max budget of 1000 steps including the data collection phase). To further understand this behavior, we analyze if the navigation policy actually looks at the target entity when the stop action is invoked. Interestingly, we find that in several instances, the target entity is visible from a slightly larger distance threshold (2.0 m) in comparison to the ImageNav policy success threshold during training (which is 1.0 m for all supplied goal frames; see appendix E.1 for details). This often happens when VLMs select goal frames in which the target entity is visible in the 2D image but not

necessarily closest in 3D space, and thus the goal frame coordinates are slightly further away than the threshold distance.

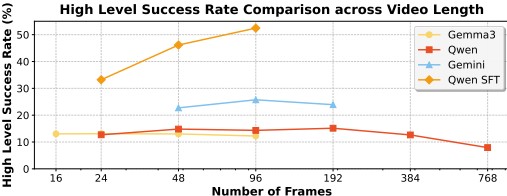 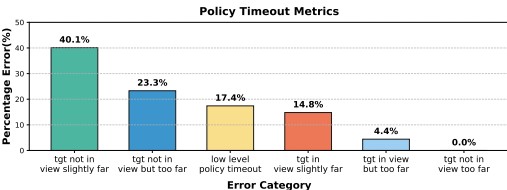

(a) Comparison across different video lengths.

(b) ImageNav Policy Errors

Figure 4: **(a)** Model performance at different number of subsampled video frames. **(b)** Reasons for ImageNav Policy Errors.

**Leveraging additional frames.** We study the effect of input length on high-level performance by evaluating agents on subsampled interaction videos of varying sizes (fig. 4a). This allows us to test whether models benefit from longer histories or merely pick up sparse visual cues. Interestingly, frozen VLMs, although capable of processing long contexts, show little to no improvement with more frames. Their performance often degrades at higher frame counts, suggesting difficulty in extracting relevant signals from densely packed, unfiltered inputs.

In contrast, the fine-tuned model (Qwen SFT) achieves clear gains when trained with longer videos. This shows that with appropriate supervision, models can move beyond shallow matching to exploit richer temporal structure in interaction histories. Nonetheless, even with full finetuning on longer (*subsampled*) videos, a large gap remains relative to maximum achievable performance (see appendix B.4). This underscores the need for future methods that enable VLMs to extract and store in a compressed manner (i.e. via memory mechanisms) task-relevant information more effectively from long video sequences within the context window.

**Cross-Benchmark Results.** In this experiment, we explore whether the spatiotemporal memory capabilities enhanced by training on simulator-based FINDINGDORY data can translate to real world settings. To evaluate this, we use VSI-Bench Yang et al. (2024), a challenging benchmark specifically designed to measure spatiotemporal reasoning in VLMs on ego-

| Method | MCQ Accuracy (↑) | Numerical Accuracy (↑) |
|---|---|---|
| Zero-Shot | 33.69 | 21.47 |
| Video-R1-Only SFT | 33.33 | 32.67 |
| Video-R1 + FINDINGDORY SFT | 35.06 | 32.35 |
| Video-R1-Only GRPO | 33.57 | 33.9 |
| Video-R1 + FINDINGDORY GRPO | **35.42** | **34.82** |

Table 3: `VSI-Bench` results. Models trained with FINDINGDORY show consistent gains.

centric real-world videos. We first augment the `Video-R1-CoT-165k` dataset introduced in Feng et al. (2025) with 40000 samples from the FINDINGDORY dataset. For effective co-training, we construct chain-of-thought traces for the entire training split of the FINDINGDORY dataset (see appendix B.5). We provide specifics about training and evaluation in appendix F. We then evaluate the trained model on VSI-Bench, showing the results in table 3. We observe that co-training with FINDINGDORY improves the SFT baseline performance by 1.7% (MCQ accuracy). Furthermore, RL-finetuning with GRPO using combined data from the FINDINGDORY dataset improves performance with a 1.85% increase over the Video-R1-Only RL checkpoint (MCQ accuracy). Overall, the consistent gains obtained from co-training point to a sim-to-real transfer effect, suggesting that FINDINGDORY can augment scarce real-world data for advancing spatiotemporal understanding in VLMs.

**Qualitative Examples.** We present multiple example responses from various VLMs in appendix H. Overall we observe that closed models perform better than open-source counterparts on single-goal tasks (see figs. 21, 24 and 25) but all models suffer on multi-goal tasks (see fig. 26). We also provide example agent video trajectories at `https://findingdorybenchmark.github.io/`.

## 6 CONCLUSION

We introduced FINDINGDORY, a benchmark for evaluating long-horizon memory in embodied agents within a highly photorealistic indoor simulation. Our findings reveal limitations in VLM context scaling, particularly in handling long observation sequences required for memory-intensive tasks. FINDINGDORY provides informative metrics that disentangle different aspects of memory reasoning, allows for procedural task generation, and efficiency-focused failure analysis beyond simple frame selection accuracy. Its extensibility allows for longer temporal dependencies, sup-

porting research on context scaling and memory efficiency in multimodal models. Our experiments also highlight challenges with hierarchical policies motivating the need for better architectures that closely integrate long-context VLM capabilities with generalist embodied policies.

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

# Supplementary Material

## A LIMITATIONS

While FINDINGDORY provides a rigorous evaluation framework for long-horizon memory in embodied AI, it has certain limitations that highlight areas for future improvement.

First, our most competitive baseline, which integrates a VLM with a navigation policy, is constrained by its low-level controller. Current memory-guided navigation policies struggle to take the shortest paths in large environments after reasoning over long interaction histories. As a result, models perform poorly on spatial awareness metrics, making it difficult to assess their true memory capabilities beyond basic path efficiency. Future work should explore improved memory-driven navigation policies to better leverage long-term spatial reasoning.

Second, in the experience collection phase, our scripted oracle policy for object interactions (Magic Grasp) executes pick-and-place actions in an unnatural way—objects are abruptly transferred from the receptacle surface to the agent's gripper without smooth motion. This may confuse VLMs, as the sudden change is visually subtle and may be overlooked if the model does not attend to the specific frames where it occurs. Developing more naturalistic interaction models could improve the realism and interpretability of memory traces.

Third, while we introduce distractor objects to discourage guessing, we do not explicitly control how prominently these objects appear in the experience collection phase. Currently, objects the agent interacts with are often viewed more frequently and closely compared to distractors, which tend to be featured less prominently in the agent's field of view. Although some tasks mitigate this by explicitly referencing non-interacted objects, future iterations of the benchmark may need stronger distractions by ensuring more balanced exposure between distractors and interacted objects.

Finally, while the experience collection phase involves both navigation and manipulation, the current interaction phase tasks are all navigation-based, as memory reasoning is primarily required for locating objects rather than executing pick-and-place actions, which are Markovian once the goal is known. However, our benchmark is easily extensible to include manipulation-based memory tasks, which would further enrich the evaluation of long-term reasoning in embodied AI.

Despite these limitations, FINDINGDORY establishes a strong foundation for studying long-horizon memory, offering extensible challenges that will evolve alongside advances in vision-language models and embodied learning.

## B BENCHMARK DETAILS

### B.1 FINDINGDORY EPISODE CREATION

In this section, we give details about our episode creation pipeline that allows us to procedurally generate data for our benchmark.

**Object-Receptacle Pairing.** The first step in the episode creation involves generating valid spawning positions for various object-receptacle pairs by sampling entities from the respective split in consideration (`train/val`). The objective is to identify *candidate objects* that can be picked up from a *start receptacle* and placed stably on a *goal receptacle*. We use the pipeline proposed in Yenamandra et al. (2023b) to procedurally generate various object placements on receptacle surfaces by running physics checks for stable placements. To make the generation procedure easier, we assume that each object is picked and placed only once during an episode. Additionally, we constrain the *start receptacle* category to be distinct from the *goal receptacle* category. We also ensure that for each *candidate object* category, a corresponding *non-interacted object* of the same category is spawned to serve as distractor entities during the interaction phase. Figures 5b and 5c provide a distribution of distractor objects and receptacles in the val episodes. The number of sampled object-receptacle pairings is equal to the number of rearrangements in the particular episode and can be configured through an argument. In essence, this controls the distribution of video frame lengths in the FINDINGDORY tasks(see Figure 5a).

**Object Placement Sequences.** Once we have identified the set of *candidate objects* and the corresponding *goal receptacles*, we require an oracle policy through which we can "naturally" drop ob-

jects on the receptacle surfaces. In practice, one can achieve this by training a "place" skill through end-to-end reinforcement learning or use a heuristic agent that can place objects on receptacles but both frameworks do not guarantee perfect $100\%$ success which is crucial for solvability in FIND-INGDORY tasks. To address this, we construct an offline pipeline that produces the set of low-level "oracle" joint actions that lead to stably placing the object on the goal receptacle. We spawn the agent (with the object snapped in its end-effector) at the associated viewpoints of *goal receptacles* on the navigation mesh. From each viewpoint, we try to execute a heuristic, state-machine based placing policy which performs a set of predefined joint effector actions after identifying a "clutter-free" region in the receptacle pointcloud (using an onboard depth sensor). The output of this routine produces a set of low level joint actions and the corresponding receptacle viewpoint for each *goal receptacle* that was identified in the previous object-receptacle pairing stage. We reject episodes where the heuristic-based place policy cannot stably place the object from *any* of the associated goal receptacle viewpoints.

**Task Creation.** Once we have the final filtered set of episodes, we populate task instructions for each episode using `yaml`-based instruction templates as used in Szot et al. (2024). We employ the instruction templates to *procedurally* populate various placeholders such as object names, attributes and appearance orders that create specific task instructions grounded in the entities (objects and receptacles) present in the episode in an automated manner. We present the diverse instructions templates we use for each task category in Table 5. In total, we create $82174/5876$ unique task instructions within the `train/val` split respectively.

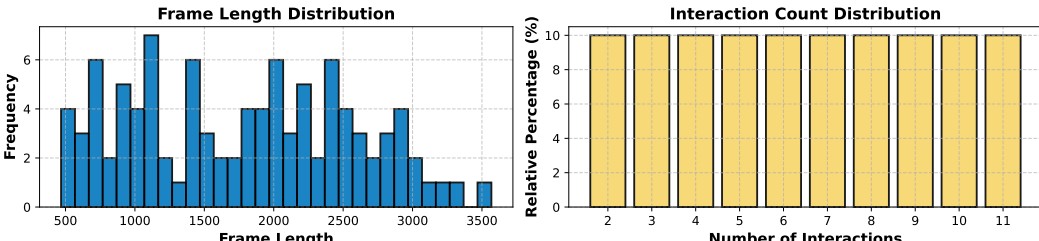

(a) Distributions of frame counts and interaction counts across episodes.

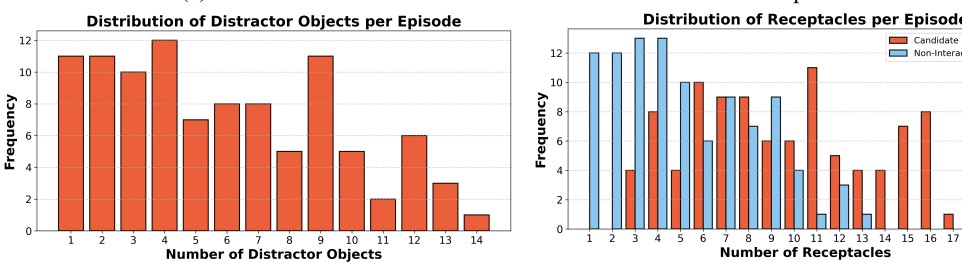

(b) Number of objects per episode.          (c) Number of receptacles per episode.

Figure 5: Summary statistics of episode data.

## B.2 EXPERIENCE COLLECTION PHASE

We now outline the details of the experience collection phase grounded in the episodes created by the pipeline outlined in the previous section. The experience collection phase is designed to collect a "clean" and "natural" video sequence of an agent interacting in an home environment while interacting with various entities. We focus on *object rearrangements* as the core interaction routine followed by the embodied agent. We construct an `OracleAgent` agent that uses privileged information from the simulator instance to efficiently generate such interaction routines. We now detail the `OracleAgent` policy to generate such video sequences for a particular episode.

**Nav-to-Pick.** The agent begins by being spawned at a random location in the scene associated with a particular episode. We then query a `ShortestPath` function that leverages privileged simulator information to navigate to the object that has to be picked up. The `OracleAgent` follows the shortest path to the object and collects images as it navigates in the environment.

**Pick.** Once the `OracleAgent` reaches the object location, it executes a hardcoded pick policy sequence which results in the object being magically grasped by the `OracleAgent`. The pick policy uses the onboard depth sensor to orient itself with the object instance placed on the receptacle

infront of the agent. Once the object is in clear view (using the ground truth instance mask from the simulator), the end effector extends outwards to perform a `magic_grasp` action. In cases when the object is not visible, we re-query the `ShortestPath` function to navigate to an alternate viewpoint associated with the receptacle that would lead to the object being clearly visible.

**Nav-to-Place.** After grasping the object, the `OracleAgent` navigates to the corresponding *goal receptacle* that was sampled during the episode creation phase. Specifically, the agent navigates to the specific viewpoint associated with the *goal receptacle* from which the pre-recorded placement action sequence is to be executed.

**Place.** Once the agent reaches the specific viewpoint, we re-play the end-effector action sequence that was found to lead to a stable object placement during episode creation. This marks the end of a single rearrangement interaction routine and the agent transitions to rearrange the next *candidate object* in the episode until no more objects are left.

**Final Navigation.** At the end of all interaction routines within an episode, the agent navigates to a predefined location to ensure that it does not remain in close proximity to the final interacted object or the final *goal_receptacle*. This ensures that tasks are not solvable by selecting only the last few frames in the video sequence. We ensure that the agent navigates to a distance of 3 m away from the final interaction location.

The above `OracleAgent` routine enables us to procedurally create videos of the agent rearranging objects in the environment. Additionally, as the agent navigates between the pick and place location, it collects additional frames detailing the scene environment which can be useful (or serve as distractor frames) when attempting to solve FINDINGDORY tasks. We present an example experience collection trajectory and associated "magic" pick-place sequence in Figures 19 and 20.

### B.3 EVALUATIONS

In this section, we define the primary evaluation metrics employed to benchmark the performance of various baselines on the FINDINGDORY tasks. Following Batra et al. (2020), we focus on metrics that systematically quantify both the success and efficiency of the agent in solving specified instructions. Success focuses on accurately selecting frames from the interaction video that lead to task completion, while efficiency evaluates if the agent selects and navigates to the optimal frame that solves the task using the fewest low-level control actions. Notably, optimizing efficiency poses a significant challenge for current methods (Ramakrishnan et al., 2024; Yang et al., 2024).

**High-Level Policy Success Rate** (`HL-SR`). This metric measures the percentage of episodes in which the high-level agent correctly predicts frame indices required to solve the specified task. We define an episode as successful if the frame(s) predicted by the VLM satisfy the following criteria:

1. *Distance-to-Goal*: The agent is within a specified distance of the target entity upon reaching the predicted frame. We use thresholds of 2.0 m for objects and 0.1 m for receptacles. The larger threshold for objects accounts for the possibility of objects being positioned on elongated receptacles (e.g., couches), potentially affecting reachability.
2. *Angle-to-Goal*: The agent is oriented within a specified angular distance toward the target entity's center. We set angle thresholds of $45°$ for objects and $90°$ for receptacles due to their typically larger dimensions.
3. *Semantic Coverage*: The semantic mask of the target entity covers at least $0.1\%$ of the pixels in the frame.
4. *Room Region Check*: For room visitation tasks, we verify if the agent is located within the designated target region.

These thresholds and criteria were qualitatively validated through manual inspection to ensure their appropriateness; During evaluation, these criteria are verified by teleporting the robot to the pose associated with the frame(s) selected by the high-level agent.

**Low-Level Policy Success Rate** (`LL-SR`). This metric measures the percentage of episodes where the low-level navigation agent successfully navigates to the target entity. The low-level agent is activated only if the high-level agent correctly predicts the frame indices, meaning `LL-SR` is evaluated conditionally upon `HL-SR` being true for each episode. The success criteria match those outlined for `HL-SR`.

**Success-weighted-by-Path-Length**. This metric evaluates task completion efficiency, computed per task episode as follows (Batra et al., 2020):

$$\mathrm{SPL}_i = S_i \cdot \frac{l_i}{\max(p_i, l_i)}$$

where $S_i$ denotes `HL-SR`/`LL-SR` for computing the `HL-SPL`/`LL-SPL` respectively, $l_i$ is the length of the shortest possible path to the closest successful goal frame (or shortest path to revisit all subgoals), and $p_i$ is the actual path length traveled by the agent.

Additionally, we introduce two relaxed success metrics to analyze baseline failure modes:

1. **Distance-to-Goal-Only Success Rate** (`DTG-SR`). Computed identically to `HL-SR` but without the semantic coverage requirement. This metric helps quantify instances where the agent selects frames close to the goal without necessarily capturing it visually.
2. **Semantic Coverage Success Rate** (`SC-SR`). Computed identically to `HL-SR` but without the distance-to-goal requirement. This metric evaluates instances where the agent correctly selects frames showing the target entity but fails to meet proximity criteria.

**FINDINGDORY receptacle navigation success criteria.** The success criteria of 0.1m for navigating to receptacle is based on the strategy prescribed in Yenamandra et al. (2023a):

1. During data generation (see appendix B), we first sample a set of candidate viewpoints within a 1.0 m radius around each receptacle. These are simply locations from which the receptacle is visible and reachable; they are stored in the episode metadata and used only as reference points.
2. At evaluation time, the actual success threshold is 0.1 m: the robot must navigate to within 0.1 m of any of these stored viewpoints to be counted as successful. Thus, the 1.0 m radius defines the region where reference points are generated, while 0.1 m is the true success criterion.

**A note on success criteria for evaluated baselines.** The above success criteria thresholds are strictly followed for all FINDINGDORY task evaluations. We highlight that these navigation success criteria can be different from success criteria that are used in specific baselines that are evaluated on the FINDINGDORY task suite. For instance, in the low-level image-goal navigation policy training, we use a navigation success criteria of 1.0 m (see appendix E.1). But a baseline that employs such a policy as the low-level agent would still only be successful when FINDINGDORY navigation success criteria are satisfied.

### B.4 SUBSAMPLED VIDEOS SOLVABILITY ANALYSIS

In this section, we analyze the solvability of the tasks proposed in FINDINGDORY suite when all videos are subsampled to enable full-finetuning of VLMs (see appendix D.5 for details). Specifically, we conduct a solvability analysis at the 96-frame video subsampling level. Full finetuning of the VLM on these videos enables highest task performance of $\approx 52\%$ as shown in fig. 4a.

From fig. 6 we observe that after subsampling the videos to 96 frames, there is a $\approx 1\%$ drop in solvability across the various low-level task categories. The largest drop is observed in the *Conditional Interaction* tasks where a specific object or receptacle is to be revisited based on the original interaction sequence. We want to highlight that this solvability calculation is solely based on whether a *solution frame* (from multiple possible frames in the full length video) is left over in the subsampled video. Such

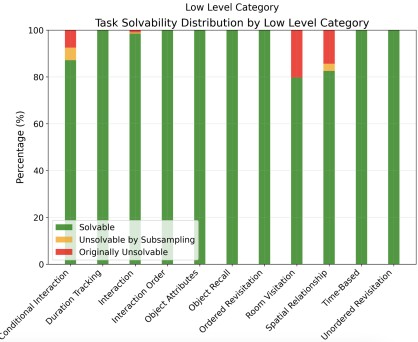

Figure 6: Task Solvability Ratio at the 96 frame subsampling level.

frames are treated as a solution based on whether they satisfy the PDDL goal conditions (see section 3.2) corresponding to the task. But such frames are extremely difficult to select for goal completion for a VLM since the subsampling routine causes a large drop in the relevant context that is needed to make sense of the solution frame. For instance, consider the task *"Navigate to the object which you interacted with which is the farthest from your current location"*: it is possible that a

frame with the target object is left over in the subsampled video. But selecting this solution frame is nearly impossible since the subsampling to 96 frames eliminates all the spatiotemporal context that is necessary to solve this task.

From fig. 6 we also observe that $3.85\%$ of the tasks are *Originally Unsolvable* which correspond to tasks focused on distractor entities such as *"Navigate to a soft toy that you did not rearrange yesterday"* or *"Navigate to a room that you did not visit yesterday"*.

## B.5 Chain-of-Thought Trace Generation

```
        Full Video Textual Memory Trace                    48-frame Subsampled Video Textual Memory Trace
"{                                                    "{
  'current_room_name': 'entryway',                      'current_room_name': 'entryway',
  'current_agent_action': 'navigating',                 'current_agent_action': 'navigating',
  'current_object_being_manipulated': 'None',           'current_object_being_manipulated': 'None',
  'current_receptacle_being_manipulated': 'None',       'current_receptacle_being_manipulated': 'None',
  'frame_indices': [0, 0],                              'frame_indices': [0, 0],
  'timestamps': ['06:00', '06:00']                      'timestamps': ['06:00', '06:00']
}",                                                   }",
"{                                                    "{
  'current_room_name': 'living room',                   'current_room_name': 'living room',
  'current_agent_action': 'navigating',                 'current_agent_action': 'navigating',
  'current_object_being_manipulated': 'None',           'current_object_being_manipulated': 'None',
  'current_receptacle_being_manipulated': 'None',       'current_receptacle_being_manipulated': 'None',
  'frame_indices': [1, 51],                             'frame_indices': [1, 5],
  'timestamps': ['06:00', '06:12']                      'timestamps': ['06:00', '06:12']
}",                                                   }",
...                                                  ...
...                                                  ...
"{                                                    "{
  'current_room_name': 'bedroom',                       'current_room_name': 'bedroom',
  'current_agent_action': 'picking_up_object_from_receptacle', 'current_agent_action': 'picking_up_object_from_receptacle',
  'current_object_being_manipulated': 'laptop_cover',   'current_object_being_manipulated': 'laptop_cover',
  'current_receptacle_being_manipulated': 'bed',        'current_receptacle_being_manipulated': 'bed',
  'frame_indices': [83, 115],                           'frame_indices': [9, 11],
  'timestamps': ['06:19', '06:27']                      'timestamps': ['06:19', '06:27']
}",                                                   }",
...                                                  ...
"{                                                    "{
  'current_room_name': 'hallway',                       'current_room_name': 'hallway',
  'current_agent_action': 'placing_object_on_receptacle', 'current_agent_action': 'placing_object_on_receptacle',
  'current_object_being_manipulated': 'laptop_cover',   'current_object_being_manipulated': 'laptop_cover',
  'current_receptacle_being_manipulated': 'bench',      'current_receptacle_being_manipulated': 'bench',
  'frame_indices': [142, 220],                          'frame_indices': [15, 22],
  'timestamps': ['06:34', '06:52']                      'timestamps': ['06:34', '06:52']
}",                                                   }",
...                                                  ...
...                                                  ...
"{                                                    "{
  'current_room_name': 'hallway',                       'current_room_name': 'hallway',
  'current_agent_action': 'navigating',                 'current_agent_action': 'navigating',
  'current_object_being_manipulated': 'None',           'current_object_being_manipulated': 'None',
  'current_receptacle_being_manipulated': 'None',       'current_receptacle_being_manipulated': 'None',
  'frame_indices': [449, 467],                          'frame_indices': [46, 47],
  'timestamps': ['07:47', '07:52']                      'timestamps': ['07:47', '07:52']
}"                                                   }"
```

Figure 7: Textual Memory Trace corresponding to agent interaction during *Experience Collection*. The memory trace is subsampled to ensure alignment with the 48-frame video.

In this section, we outline the pipeline that is employed to generate detailed chain-of-thought traces for the FindingDory training data split. We begin by generating textual memory traces that detail the interactions performed by the robot during the *Experience Collection* phase (see Appendix B.2). For this, we track all the individual actions that are executed by the robot along with necessary information such as time of day, object-recetpacle info, room names that are necessary for successful task completion. For the co-training experiments for cross-benchmark analysis, we subsample all videos to 48 frames and accordingly modify the textual memory traces to ensure alignment between the memory trace and subsampled video frames. A sample original and subsampled textual memory trace is shown in Figure 7.

To generate faithful chain-of-thought traces for all samples in the training split, we leverage a zero-shot, text-based LLM to generate a rationale for each task based on the textual memory history and corresponding keyframe solution. We hypothesize that the zero-shot LLM would be able to generate more consistent traces if it is given access to the solution keyframes and prompted to

generate a rationale using the solution. This is in contrast to using the LLM to predict the solution keyframes based on the textual memory trace which is a more difficult task. We use the open-source `DeepSeek-R1-Distill-Qwen-32B` (Guo et al., 2025) for this purpose and leverage the vLLM library (Kwon et al., 2023) for accelerated inference. We use the prompt structure as shown in Figure 8. After the initial chain-of-thought trace generation, we observed that $\approx 8\%$ of the generated traces contained references to the solution keyframe list provided in the prompt. To fix these erroneous traces, we use the `DeepSeek-R1-Distill-Qwen-32B` model to rephrase the generated traces to remove any references to the solution keyframe list following the prompt shown in Figure 9. After the initial trace generation and rephrasing steps, we are left with $7.8\%$ traces in which the solution keyframes list predicted by the LLM contains a keyframe which does not belong to the actual solution keyframes list. All these samples are discarded which leads to a total of 72973 training samples with labeled chain-of-thought traces.

---

### `DeepSeek-R1-Distill-Qwen-32B` CoT Generation Prompt

You are an intelligent agent assisting a robot in completing specific goals by analyzing interactions recorded during its previous navigation around a house. The interaction history captures the robot rearranging objects (pick and place). The robot is given a specific task that needs to be solved by selecting a specific frame index between 0-48. Each task is solved by selecting the correct frame from the history.

Your task is to help the robot generate a detailed reasoning chain of thought to solve the task. To aid in generating the reasoning trace, you will be given the full list of oracle keyframes that will solve the task. Choosing any frame from this list solves the task. If the oracle list contains only [-1] element, then the task is just unsolvable. The reasoning_trace should terminate with a list of predicted solution keyframes. Use the oracle solution to verify your solution_ keyframes predicted list.

You need to solve the following task: {task_instruction}.
The oracle solution is: {oracle_solution}.
The detailed interactions of the robot are: {textual_memory_trace}

Explanation of oracle_solution list format:
- Single list with multiple sublists. Each sublist can contain multiple frame indices.
- If only one sublist, then the task is a single-goal task
- If multiple sublists, then the tasks has multiple subgoals and each sublist has corresponds to the solution frames of respective subgoal

Please generate the detailed reasoning trace and make sure you refer to the various frame indices and robot actions in the interaction history.

Guidelines for reasoning_trace generation: - Do not include explicit references to the oracle keyframes.
- Your answer should always end with the list of frames that solve the task (use [-1] if unsolvable) based on the reasoning trace.
- Ensure the predicted solution_keyframes follows the format of the oracle_solution list.
- Make sure that each value in the final list of solution_keyframes you generated exists in the oracle_solution list that is provided.

Please follow the output format: Output Format (should be in JSON format):

```
{
  "reasoning_trace": str.
  "solution_keyframes": str.
}
```

Figure 8: Prompt used for CoT trace generation with `DeepSeek-R1-Distill-Qwen-32B`.

### B.6 SANITY CHECKS

We investigate how a VLM would perform at short-horizon visual comprehension on images from the same episodes used for long-horizon evaluation in our benchmark. This tests if poor performance of the evaluated models on our long-horizon reasoning tasks is due to a domain or sim2real gap. We assess this by sampling frames around object interactions, asking models to list out detected objects, and check accuracy compared to the ground truth category via string matching – a conservative

---

### DeepSeek-R1-Distill-Qwen-32B CoT Trace Rephrasing

You are given a task instruction and a reasoning trace (chain of thought) that was generated to solve the task. The reasoning trace currently contains explicit reference to "oracle solution list" or similar oracle-based information that should not be visible to a model trying to solve the task independently.

Your job is to rephrase the reasoning trace to remove any explicit references to oracle solutions or oracle keyframes, while preserving the core reasoning logic and thought process. The rephrased reasoning should read as if it was generated by a model thinking through the problem step by step without access to ground truth answers.

You should keep references to all other details such as individual frame indices or segments intact as they are necessary for reasoning.

Task Instruction: {task_instruction}
Original Chain of Thought: {existing_chain_of_thought}
Please follow the output format: Output Format (should be in JSON format): "'

```
{
    "reasoning_trace": str.
}
```

Figure 9: Prompt for CoT Trace Rephrasing with DeepSeek-R1-Distill-Qwen-32B.

metric that misses correct predictions using synonyms or alternative phrasings. This still results in image sequences of length 50-150, but Gemini achieves an accuracy of 67% which is significantly higher than it's average high level success rate across tasks in fig. 2. This shows that despite clear, close-up views, models struggle to reason over them when temporal sequence comprehension is required.

## C EXTENDED RELATED WORK

### C.1 AN OVERVIEW OF APPROACHES TACKLING MEMORY IN LONG VQA AND EMBODIED TASKS

Many recent works equip large language models with memory-like capabilities through task-specific knowledge bases or semantic maps for navigation planning (Min et al., 2024; Chang et al., 2023). MobilityVLA (Xu et al., 2024) leverages long-context VLMs to process past frames, enabling goal-frame selection via Gemini, while topological mapping guides navigation without explicit pathfinding. ReMEmbR (Anwar et al., 2024) and Embodied RAG (Xie et al., 2024) introduce nonparametric memory trees to store and retrieve past experiences for planning. For video question answering, Zhang et al. (2023); Kahatapitiya et al. (2024) propose a streaming-based approach that accumulates frame-wise captions for later querying. However, this method constrains memory to the expressivity of vision-language or captioning models, potentially omitting critical details. We include the hierarchical approach involving long-context VLMs and the approach utilizing textual memory traces as baselines in our evaluations on the benchmark tasks.

## D BASELINE DETAILS

### D.1 GEMINI AGENT

We experiment with the `gemini-2.0-flash` model which can process upto 1 million tokens in the context window. We strictly follow the outlined Google Gemini API developer guidelines (gem) to structure the prompt and perform API calls over the long videos in the FINDINGDORY tasks. We do not use a "structured output" format such as json in the output responses as we empirically observed it failed to produce coherent outputs when supplied with long videos in the context. Since the `gemini-2.0-flash` model expects the input frames to be sampled at 1 FPS, we prompt the model to directly generate specific timestamps in the recommended `MM:SS` format to localize the frame corresponding to the target entity. We provide the detailed evaluations prompt in Fig. 10.

---

### gemini-2.0-flash Task Prompt

You are an intelligent question answering agent. In the prompt video preceeding the text, you will be shown a long stream of images that have been collected by a robot while navigating around a home and asked to answer a question about the space to assist a user in completing tasks in this home. In the video stream, the robot is picking up objects and rearranging them to different arbitrary locations throughout the house.

The questions asked by user will require being able to look at the full set of images collected by robot to be able to provide the answer. Your task is to identify the exact point in the video (by timestamp) where the user should move to best accomplish their goal. You can do this by outputting the exact timestep in the video where you are most confident that the object or place that they should move to, was viewed closely.

The images in the video also have the time of day information in top left. You need to use the time of day information for tasks that require to revisit objects or receptacles at a specific time of day.

For some tasks that require revisiting multiple receptacles or objects, you should output multiple frame indices corresponding to the correct order of revisitation (only if specified). For all user goals, identify the minimum number of objects or receptacles (targets) that you need to revisit to complete the user's goal. For each revisitation, you should select one corresponding timestep in the video that you will move to. You need to ensure that you are not revisiting targets that are not relevant to the user's goal.

Generate timestamps in the MM:SS format where the first two digits represent minutes and the last two digits represent seconds. To solve the tasks effectively, you should try to summarize the video by listing out the objects that were picked and placed, the receptacles from which each object was picked up and the receptacle where it was dropped.

The user's goal is to: {goal}. Now look at the full video containing all the images. Give your final response with NUM_TARGETS_TO_REVISIT: ||<num_target>||. TIMESTAMP_INDEX: ||<timestamp_1>,...,<timestamp_num_target>||. You can include multiple timestamps if the task requires visiting more than one target (one timestamp for each target). Do not add any other text.

Figure 10: Prompt for gemini-2.0-flash model

## D.2 QWEN, GEMMA AND GLM AGENT

We run the Qwen2.5-VL-7B (Bai et al., 2025), Gemma3-12B (Team et al., 2025) and GLM-4.1V-Thinking (Hong et al., 2025) locally for evaluations on the FINDINGDORY tasks. The Qwen model uses a maximum of 768 frames sampled uniformly across the input sequence while Gemma and GLM does not impose a maximum frame limit. For evaluation on 768 frames, we use two A40 GPUs with 48GB of memory. For experiments with shorter context lengths and higher subsampling (24–384 frames), a single A40 GPU suffices. We provide the full task prompt we use for inference with the three agents in Fig. 11 and utilize structured JSON outputs.

## D.3 GPT AGENT

We use the official OpenAI GPT-4o (Hurst et al., 2024) API to run evaluations on the FINDINGDORY tasks with video frames subsampled at 96 frames to optimize API costs. For running complete evaluation on the entire validation task suite, we incurred a total cost of $\approx 400$ USD. We provide the full system prompt used for GPT-4o evaluations in Fig. 12.

## D.4 TEXT AGENT

We implement the the text-based VLM agent using the Qwen2.5-VL (Bai et al., 2025) model. For every chunk_size frames in the video, we generate a description using the VLM. We then attach the frame indices and *time of day* information to the generated summary for the frame window in consideration. We use the prompt shown in Fig. 13 to generate the structured JSON outputs for each chunk in the video. We then concatenate all the generated JSON outputs and pass it to the Qwen model to perform text-only reasoning over the entire text-based history of the video. The text-based reasoning prompt is provided in Fig. 14. We note that using the same Qwen model for generating the individual text summaries and performing the final text-based reasoning helps in optimizing inference load as we maintain only a single instance of the VLM through the entire evaluation.

## Qwen/Qwen2.5-VL-7B-Instruct, google/gemma-3-12b-it and zai-org/GLM-4.1V-9B-Thinking Task Prompt

You are an expert and intelligent question answering agent. You will be shown a video that was collected by a robot yesterday while navigating around a house and picking and placing objects. Your job is to help the robot complete a task today by looking at the video and finding the frame indices that the robot should move to. Make sure your response is all within the JSON.

Note:
- The robot uses a magic grasp action to pick up an object, where a gripper goes close to the object and the object gets magically picked up.

Output Format (should be in JSON format):

```json
{
  "chain_of_thought": str.
  "frame_indices": list[int].
}
```

Where:
- Chain of Thought: A detailed explanation of the thought process in determining the frame indices.
- Frame Indices: The frame indices in the video stream showing the object or place that the robot should move to.
- When deciding which frame indices to choose, make sure you choose the frame indices that are closest to the object/place.
- If task requires the agent to go to multiple places, output one frame index per object/place. (Do not use ellipsis)

The robot's goal is: {goal}

Figure 11: Prompt for `Qwen2.5-VL` and `Gemma3` model

## gpt-4o-2024-07-18 Task Prompt

** OBJECTIVE **
You are an expert and intelligent question answering agent. You will be shown a video that was collected by a robot yesterday while navigating around a house and picking and placing objects.
Your job is to help the robot complete a task today by looking at the video and finding the frame indices that the robot should move to, to complete the task.

** OUTPUT FORMAT **

```json
  "chain_of_thought": str.
  "frame_indices": list[int].
```

Where:
- Chain of Thought: A detailed explanation of the thought process for determining the frame indices.
- Frame Indices: The frame indices in the video stream showing the object or place that the robot should move to.
- When deciding which frame indices to choose, make sure you choose the frame indices that are closest to the object/place.
- If task requires the agent to go to multiple places, output one frame index per object/place.

** NOTES **
- The robot uses a magic grasp action to pick up an object, where a gripper goes close to the object and the object gets magically picked up.

** TASK **
The robot's goal is: {goal}

Figure 12: Prompt for `gpt-4o` model

---

## Text Agent(VLM) Task Prompt

You are an expert and intelligent question answering agent. You will be shown a stream of `{chunk_size}` images taken by a robot while navigating around a simulated house and picking and placing objects. Your job is to describe the image based on the output format. Make sure your response is all within the JSON format.

Output Format (should be in JSON format):

```json
{
  "room_name": str.
  "picking_placing_or_navigating": str("picking", "placing" or "nav").
  "object_being_manipulated": str.
  "receptacle_being_manipulated": str.
  "other_objects_in_scene": list[str].
}
```

Where:
- Room name: The name of the room the robot is in.
- Picking, placing, or navigating: Whether the robot is picking or placing an object, or navigating around the house.
- Object being manipulated: The object that the robot is picking or placing (if relevant).
- Receptacle being manipulated: The receptacle that the robot is picking up from or placing an object into (if relevant).
- Other objects in scene: Objects present in the scene besides the main object and the receptacle.

Note:
- The robot uses a magic grasp action to pick up an object, where a gripper goes close to the object and the object gets magically picked up.

---

Figure 13: Prompt to generate text summaries for `chunk_size` frames with `Qwen2.5-VL`

### D.5 SUPERVISED FINETUNING BASELINE

We leverage the FINDINGDORY training split which consists of $82174$ unique tasks across the categories mentioned in Table 2. Each video-instruction pair is coupled with a list of ground-truth frame indices that solve the task. For multi-goal tasks, each subgoal has a corresponding ground-truth frame indices sublist. Since the FINDINGDORY video sequences can be extremely long, we uniformly subsample all videos to 96 frames. The ground truth frame indices are also subsampled to match the shorter video lengths to ensure the model chooses from within the subsampled frames.

**Training Details.** We conduct full-finetuning of the `Qwen/Qwen2.5-VL-3B` model using the open-source `huggingface/trl` library. For hyperparameter optimization, we utilized a smaller representative split of the training dataset with $36274$ samples and conducted a grid search over the learning rate, num_epochs and weight_decay parameters. The best checkpoints from all runs were selected by evaluating performance on all video-instruction pairs in the full validation split. This "offline" evaluation is conducted without instantiating the simulator by directly comparing the model's predicted keyframe indices with the ground-truth keyframe indices using exact string matching (thus, we dont compute metrics proposed in Section 4.2). We use the cosine learning rate decay scheduler for all experiments. For the final training on the larger dataset, we found that a learning rate of $5 \times 10^{-6}$ over $5$ epochs with $0$ weight decay worked best. For all experiments, we train in bfloat16 format using 8 A40 GPUs on a single node with a batch size of 1 and gradient_accumulation of $4$ - providing an effective batch_size of 32 samples. To further optimize training, we use gradient_checkpointing and highly optimized flash_attention implementation. In this setup, the full training run takes $\approx 120$ hrs to complete.

**Goal Sampling for Online Simulator Eval.** The best trained checkpoint is used in online simulator evaluations on the FINDINGDORY task suite. Since the model is trained to predict the complete list of ground-truth frame indices, we select a single representative index from each generated subgoal solution list to serve as the predicted frame index. Empirically, we found that choosing either the first or a random index (of a sublist corresponding to a subgoal) yields comparable performance,

---

### Text Agent(LLM) Task Prompt

You are an expert and intelligent question answering agent. You will be provided a list of textual descriptions of the environment created by an agent while navigating around a house and picking and placing objects. The descriptions correspond to logs made after collecting and viewing a small chunk of frames, and contain information about which frame numbers they correspond to. You will also be provided a goal that now needs to be accomplished, for which you will need to use the history to decide where to go. Your job is to identify the desired frame index to navigate to, based on the provided task. Make sure your response is all within the JSON.

Agent's History information:
- Room name: The name of the room the agent was in.
- Picking, placing, or navigating: Whether the robot was picking or placing an object, or navigating around the house.
- Object being manipulated: The object that the robot was picking or placing (if relevant).
- Receptacle being manipulated: The receptacle that the robot was picking up from or placing an object into (if relevant).
- Other objects in scene: Objects present in the scene besides the main object and the receptacle.

Output Format (should be in JSON format):
```json
{
  "chain_of_thought": str.
  "frame_indices": list[int].
}
```
Where:
- Chain of Thought: A detailed explanation of your thought process in determining the frame indices.
- Frame Indices: A list of one or more frame indices relevant to accomplishing the goal.
- If its a single object/place task, output only one frame index.
- If task requires the agent to go to multiple places, output one frame index per object/place. (Do not use ellipsis)

Agent's history: {history}

The goal is {goal}

Figure 14: Prompt to perform reasoning over the combined textual summaries with `Qwen2.5-VL`

while using the last index results in slightly worse performance. Therefore, we use the first index of each list as a simple and effective heuristic.

## E   LOW-LEVEL POLICY DETAILS

### E.1   IMAGE-GOAL NAVIGATION AGENT

**Policy Training.** We train the image-goal navigation policy using the architecture and training recipe described in Yadav et al. (2023). Specifically, we employ an end-to-end reinforcement learning policy trained using DDPPO, which predicts discrete navigation actions conditioned on visual RGB inputs and a goal image. The agent utilizes a ViT based visual encoder along with a 2-layer LSTM backbone. Similar to OVRL-v2 (Yadav et al., 2023), the output patch representations from the ViT are reshaped into a 3D grid and downsampled to a lower dimension using a convolutional layer called the compression layer. However, we slightly deviate from OVRL-v2 by first concatenating the patch representations from the current and goal images before passing them through the compression layer.

The policy training occurs in the training scene split of the HSSD dataset (Khanna* et al., 2023), consisting of 1166 episodes distributed across 111 scenes (a subset of the original 125 training scenes). In the Habitat simulator, the agent is modeled as a Hello Robot Stretch (Kemp et al., 2022) with a height of $1.41$ m and a cylindrical base radius of $0.25$ m. The agent's RGB sensor is positioned at a height of $1.31$ m, with a resolution of $160 \times 120$ pixels and a horizontal field of view of $43°$.

During training, each episode has a maximum step budget of $1000$ steps, with success defined by invoking the STOP action within $1.0$ m of the goal image position.

As our visual encoder, we use the VC-1-Base (Majumdar et al., 2023) model which was previously finetuned with ImageNav on a smaller training scene split from the HSSD dataset. Empirically, we found that freezing the visual encoder after this targeted finetuning significantly accelerates the overall training performance. During the VC-1 finetuning, we use a smaller learning rate of $1.5 \times 10^{-6}$ for the encoder.

We train the agents for a total of $500$M timesteps on 32 A40 GPUs running 32 parallel environments each (1024 envs in total). Following the approach from (Majumdar et al., 2023), we collect $64$ steps of experience and subsequently perform 2 PPO epochs with a mini-batch size of 2. The reward function employed is the improved formulation proposed by Yadav et al. (2023), using parameters: success weighting $c_s = 5.0$, angle success weighting $c_a = 5.0$, goal radius $r_g = 1.0$, angle threshold $\theta_g = 25°$, and slack penalty $\gamma = -0.002$. We optimize using AdamW with a learning rate of $2.5 \times 10^{-4}$ and weight decay $10^{-6}$.

We select the best-performing policy checkpoint based on a validation success rate obtained by evaluating on 1000 ImageNav episodes from the FINDINGDORY validation set, achieving a success rate of 78% and SPL of 48.2%.

### E.2 EXPLICIT MAPPING AGENT

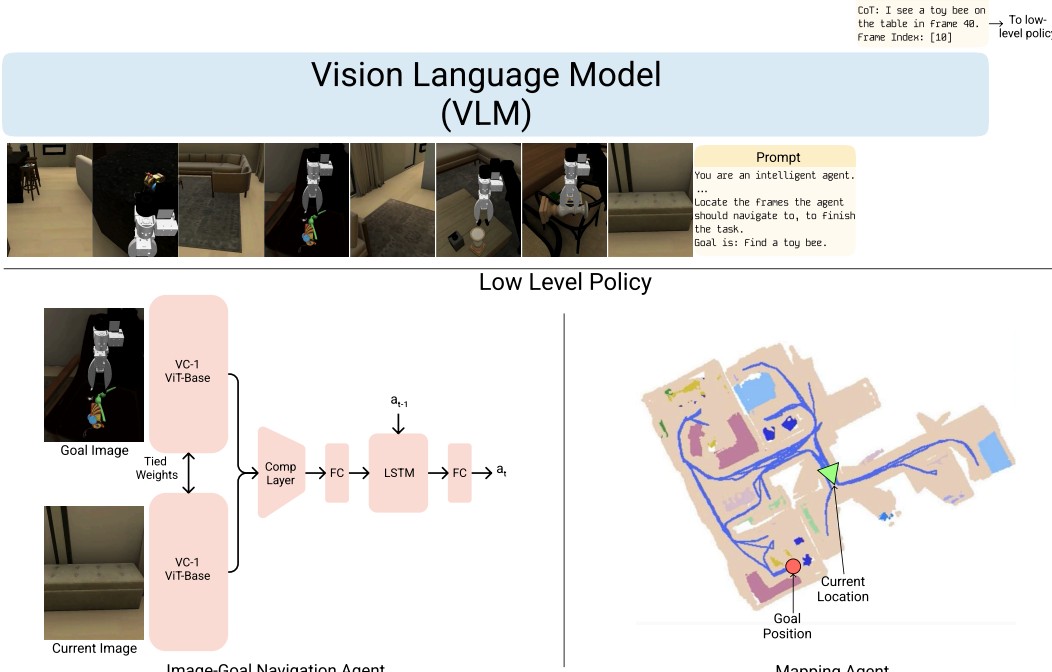

Figure 15: Hierarchical agent architecture when using a VLM for goal frame selection and low level policies for action generation.

We augment the OracleAgent (see Appendix B.2) with a top-down occupancy map as it navigates the environment to generate the video sequence. In addition, we also store the 2D poses corresponding to each frame that is collected in the video sequence. The constructed map is employed by a path planning module to generate a deterministic navigation sequence to the 2D map coordinates that correspond to the image-goal location selected by the high-level goal selection agent (see Section 4.1).

**Map Construction.** We leverage the onboard depth sensor and project the depth image at each timestep into an egocentric pointcloud using camera intrinsics. The pointcloud is then binned along the z-axis to compute occupancy values on a local 2D grid map centered around the current position of the agent. The pose sensor provides the instantaneous pose of the agent which is employed to

register the computed local map in the global map of the environment. In this way, we continuously update the global map as the `OracleAgent` navigates the environment and visits multiple pick-place targets. We freeze the map updates during the pick-place subroutines as no additional occupancy information is encountered during their execution.

**Planner.** When the high-level goal agent selects a particular goal frame, we lookup the corresponding target 2D pose in the global map. Similar to the local navigation policy employed in Chang et al. (2023), we use the Fast Marching Method (Sethian, 1996) to generate the shortest path to the goal pose. We use one of four discrete navigation actions (see Section 3.2) to reach each waypoint on the generated shortest path.

## F  CROSS-BENCHMARK EXPERIMENTS

In this section, we provide details about the training and evaluation setup for results on the `VSI-Bench` Yang et al. (2024) dataset. For evaluation, we follow the exact same evaluation protocol outlined in the original paper. In table 3, we report results across the multiple-choice (2490 in number) and numerical regression (2640 in number) QA categories, each of which has four sub-categories. For all experiments, we use the `Qwen2.5-VL-3B` as the base model. Each SFT/RL-finetuning experiment uses a single node of 8 A40 GPUs with 48 GB VRAM each.

**Supervised Finetuning.**  We conduct supervised finetuning of model with maximum of 48 frames in the input context window. We also use 48 frames during model inference as that works best between 32, 48 and 64 frame inference for all baselines. For SFT experiments with the `Video-R1-CoT-165k`-FINDINGDORY mixture, we experimented with various mixing ratios by changing the amount of FINDINGDORY data ($12.5\%, 25\%, 37.5\%, 45\%$) and found $25\%$ (40000 samples) to work best. All SFT models are trained for one epoch following the hyperparameters from Feng et al. (2025). We only evaluate the final checkpoint after one epoch for all baselines. Specifically, we use a learning rate of $1 \times 10^{-6}$, gradient_accumulation of 2, max_grad_norm of 5 and train with bfloat16 precision on 8 GPUs We leverage gradient_checkpointing, flash_attention and deepspeed_zero_2 to optimize training.

**RL Finetuning.**  For RL finetuning with GRPO, we reduce the context window to 32 frames to avoid out-of-memory errors encountered with 48 frames. Empirically, we observe that inferencing with 64 frames works better than 32 frame inference for all baselines. We mix the full FINDINGDORY training dataset ($\sim 78$k samples) with the `Video-R1-260k` dataset from Feng et al. (2025). Because each training step is computationally expensive, we train for only 1000 gradient steps using a group size of 8, batch size of 1 per GPU, and gradient accumulation of 2. As a result, the model is exposed to only $\sim 2000$ unique samples during training. We set the weight decay to 0.01, maximum gradient norm to 5, learning rate to $1 \times 10^{-6}$, and maximum completion length to 768. We evaluate checkpoints at a 100 step frequency and select the checkpoint with highest average MCQ+Numerical accuracy for each baseline.

For FINDINGDORY samples, we design a custom reward function that rewards the precision of the predicted goal frames. Let $\mathcal{R} = (R_1, \ldots, R_n)$, $\mathcal{H} = (H_1, \ldots, H_m)$, where $R_i \subset \mathbb{Z}$ (reference frame sets) and $H_i$ are finite sequences of predicted frames.

Define the score $S(\mathcal{H}, \mathcal{R})$ as

$$
S(\mathcal{H}, \mathcal{R}) = \begin{cases} 0, & \text{if } n \neq m, \\ \dfrac{\displaystyle\sum_{i=1}^{n} \sum_{h \in H_i} \mathbf{1}[h \in R_i]}{\displaystyle\sum_{i=1}^{n} |H_i|}, & \text{otherwise (assuming } \sum_{i=1}^{n} |H_i| > 0). \end{cases}
$$

## G    FAILURE ANALYSIS OF QWEN-SFT VS QWEN-ZERO-SHOT

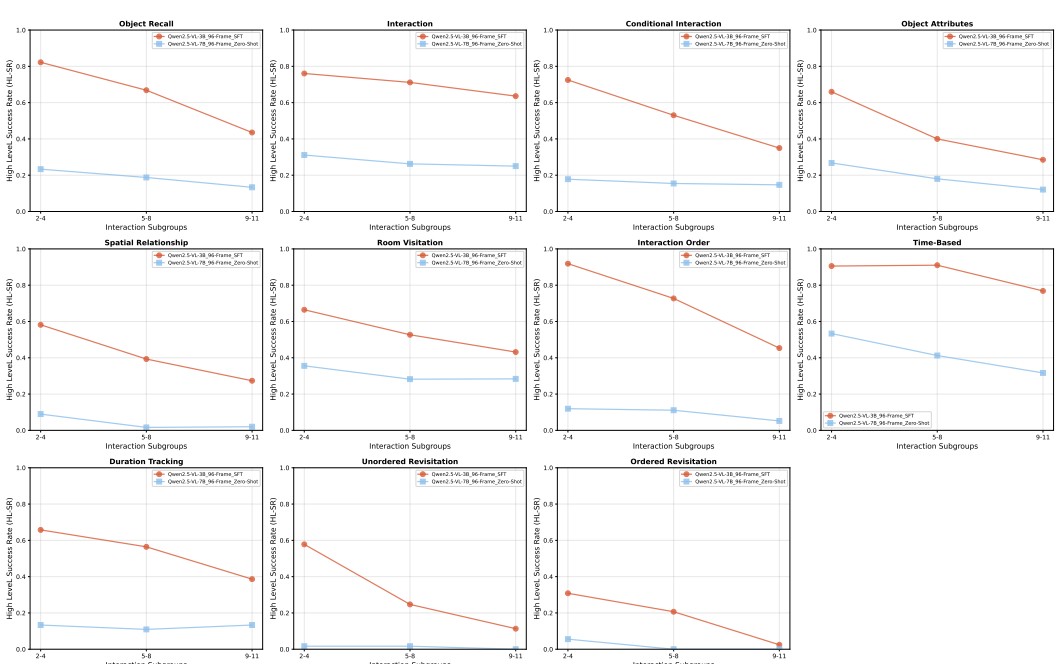

Figure 16: Success Rates of `Qwen2.5-VL-SFT` vs `Qwen2.5-VL-7B` model on various FINDINGDORY tasks grouped by the number of interaction per video.

In fig. 17, we present detailed breakdown of the High-Level Success Rate (HL-SR) of our finetuned `Qwen2.5-VL-3B` model (see appendix D.5 for details) and the zero-shot `Qwen2.5-VL-7B` baseline against the number of interactions divided in 3 subgroups. Both models are evaluated with 96 input frames. We find that the finetuned `Qwen2.5-VL-3B` consistently outperforms the zero-shot `Qwen2.5-VL-7B` across all task families. The largest relative improvements are in the 2–4 interaction regime, with categories such as Interaction Order, Unordered Revisitation, and Spatial Relationships seeing the biggest improvements. As the number of interactions increases, the performance gap narrows. At the

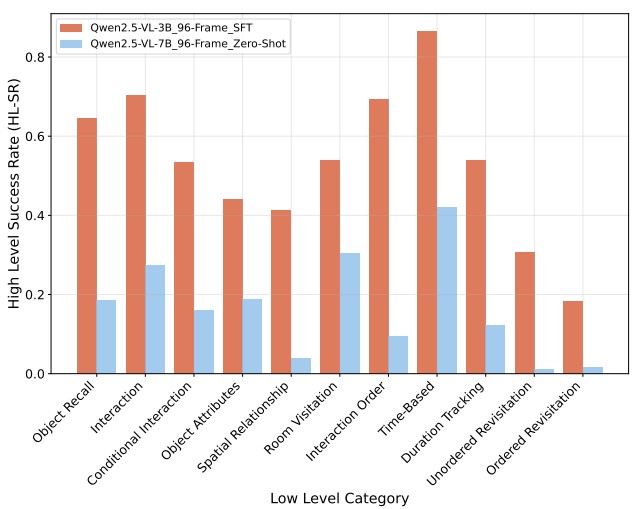

Figure 17: Average Success Rates of `Qwen2.5-VL-SFT` vs `Qwen2.5-VL-7B` across FINDINGDORY tasks.

9–11 interaction range, the smallest differences are observed for all the multi-goal tasks and the Room Visitation, and Object Attributes tasks. Overall, these trends indicate that finetuning yields the greatest benefits when given videos with more frames per second, while the advantage diminishes as trajectories become denser and more visually cluttered within a fixed 96-frame budget. This suggests that additional gains may be achievable with larger frame budgets.

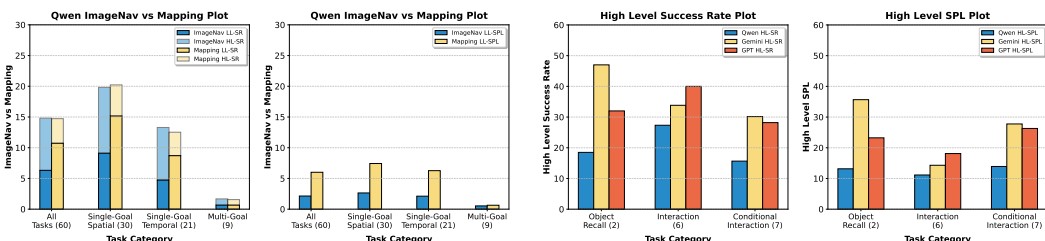

(a) Qwen ImageNav: High vs Low-Level SR & SPL.   (b) Qwen vs Gemini vs GPT: High-Level SR and SPL.

Figure 18: Extended hierarchical policy performance plots.

# H QUALITATIVE EXAMPLES

In this section, we include representative qualitative examples from the FINDINGDORY task suite. In Figure 19, we provide a birds-eye view depiction of the trajectory executed during experience collection (see Appendix B.2) in a validation episode involving 5 pick-place interaction routines. For this episode, the collected video consists of 1308 frames in total. We provide detailed responses (and meta-analysis) from various baselines across the various task categories in Figures 21 to 26. In the following, we briefly discuss the different failure modes encountered in the generated responses across the 3 high-level task categories defined in Table 2.

**Single-Goal Spatial Tasks.** We observe that for the *Object Recall* task (fig. 21), the models are able to identify a frame with the target *chest of drawers* but do not select the closest frames to ensure success. In the *Interaction* based tasks (figs. 21 and 22), we observe models fail to accurately list the sequence of interaction events which can be partially attributed to supplying only 96 subsampled frames from the original video. In one case, GPT-4o instead focuses on a "distractor" target object (fig. 22). All models perform poorly on the *Spatial Relationship* tasks (fig. 22) as they are unable to coherently reason about the spatial layout of the house in the provided video.

**Single-Goal Temporal Tasks.** For the *Interaction Order* task, we observe that the closed-source models are able to accurately identify the target object but the open-source variants fail to do so (fig. 24). Models are generally able to select the correct frame in the *Time-Based* task as the model only needs to retrieve a single frame corresponding to the correct overlaid timestamp. For the *Duration Tracking* task, only GPT-4o generates the correct reasoning and frame prediction. Intersetingly, Gemma-3 tries to track the duration spent in each room by matching the floor color observed in the images but eventually fails at the task (fig. 25).

**Multi-Goal Tasks.** All models fail to perform successfully on tasks under this category. Only GPT-4o gets a correct response for all subgoals. Surprisingly, the reasoning trace has a hallucination but this mistake is omitted in the final frame sequence prediction (fig. 26). The models struggle to coherently list the event sequence and also hallucinate interactions. Furthermore, Gemini-2.0-flash also struggles to adapt the sequence of events it lists into the query revisitation sequence specified in the task (fig. 26).

| Method | All Tasks | Object Recall | Inter-action | Conditional Interaction | Object Attributes | Spatial Relation | Room Visitation | Interaction Order | Time-Based | Duration Tracking | Unordered Revisitation | Ordered Revisitation |
|---|---|---|---|---|---|---|---|---|---|---|---|---|
| Oracle | 95.92 | 100.00 | 98.67 | 92.03 | 99.80 | 85.42 | 80.00 | 99.86 | 99.50 | 100.00 | 99.83 | 99.00 |
| Text Agent | 9.45 | 9.50 | 20.00 | 7.99 | 6.84 | 4.17 | 19.36 | 7.37 | 24.50 | 15.46 | 0.83 | 0.67 |
| Gemma3 | 13.04 | 19.50 | 9.00 | 13.22 | 17.82 | 5.83 | 30.61 | 12.31 | 28.50 | 21.13 | 0.33 | 2.00 |
| Qwen2.5-VL | 15.14 | 15.50 | 35.00 | 15.22 | 17.62 | 4.58 | 31.83 | 9.83 | 40.00 | 12.76 | 1.83 | 1.33 |
| GLM-4-1v | 23.49 | 29.00 | 49.17 | 24.59 | 30.35 | 7.08 | 36.34 | 21.05 | 50.50 | 16.11 | 3.33 | 3.00 |
| Gemini-2.0-Flash | 25.73 | 47.00 | 33.83 | 30.16 | 35.27 | 10.42 | 41.01 | 23.23 | 43.50 | 21.48 | 8.50 | 9.33 |
| GPT-4o | 27.33 | 32.00 | 40.00 | 28.19 | 25.20 | 14.58 | 44.68 | 28.56 | 63.00 | 24.83 | 6.33 | 7.00 |
| Qwen SFT | 52.44 | 64.50 | 70.63 | 46.14 | 42.65 | 24.58 | 53.47 | 68.22 | 88.50 | 53.53 | 29.70 | 20.00 |

Table 4: High-level goal success rate (%) across task categories. The *Oracle* row is shaded in gray.

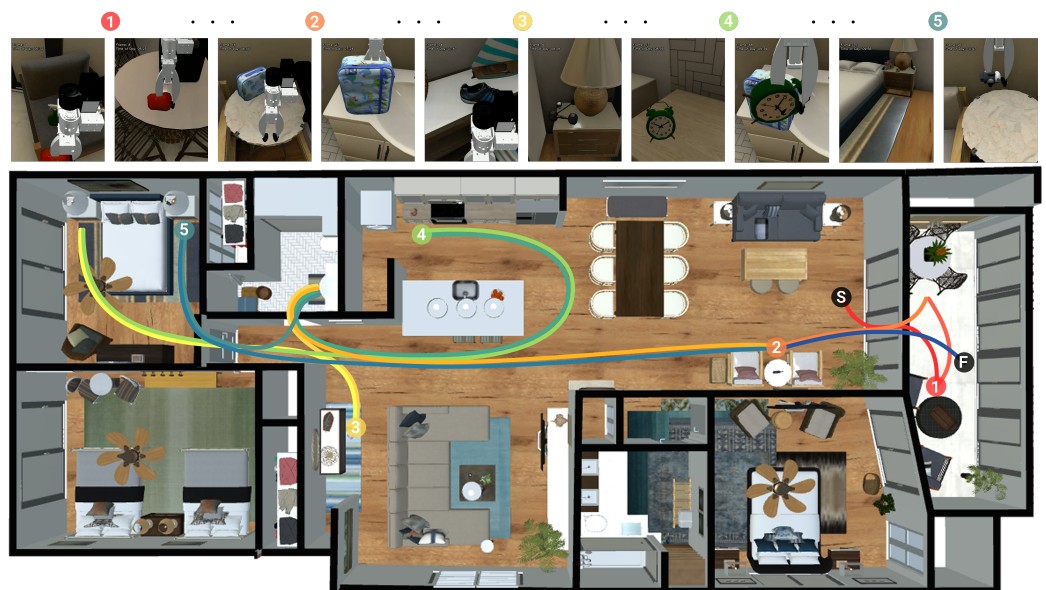

Figure 19: An example experience collection trajectory executed by `OracleAgent`.

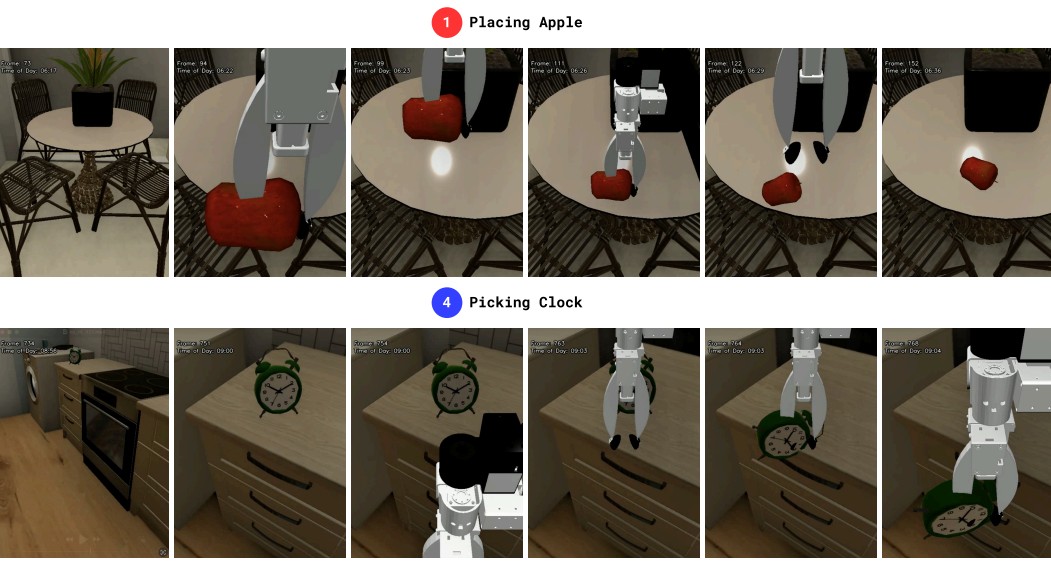

Figure 20: Example "magic" pick-place sequences executed by the `OracleAgent` for sample episode in Fig. 19.

## Single-Goal Spatial Tasks

### Object Recall

**Task:** Navigate to a chest of drawers.

✗ **GPT-4o:** To accomplish the task of navigating to a chest of drawers, I need to identify the frames where the chest of drawers is visible. The focus will be on frames showing the chest of drawers most prominently and closely. Starting from frame 0, I will scan through the images focusing on furniture that matches a typical chest of drawers. I observed a chest of drawers in frames 53 and 54 located in the kitchen area, which confirms the presence in those frames.
**frame_indices:** [53]
Meta Analysis: Misidentifies kitchen counter as target object

✓ **Gemini-2.0-flash:** The robot performs the following actions: - Picks up a green spray bottle and blue lunchbox from a chair and places it on a table. - Picks up an apple from a table and places it on another chair. - Picks up the lunchbox from the table and puts it in the bathroom sink. - Picks up a shoe from a shelf and places it next to a lamp on a bedside table. - Picks up a green alarm clock from a dresser and places it near the washing machine. - Picks up a panda figurine from a shelf and places it next to the lamp. - Picks up the spray bottle from the bedside table. - Picks up the panda figurine from bedside table and places it near a chair. The user wants to navigate to a chest of drawers. The video shows chest of drawers at [00:37], [00:47], [00:53], [00:71], [00:78]. NUM_TARGETS_TO_REVISIT: ||1||. TIMESTAMP_INDEX: ||00:37||
**frame_indices:** [37]

✗ **Qwen-2.5-VL:** The robot needs to navigate to a chest of drawers. In the video, the robot moves through various rooms and eventually reaches a room with a chest of drawers visible on the right side of the screen.
**frame_indices:** [49]
Meta Analysis: Selects a frame too distant from the target despite a closer one being available

✗ **Gemma-3:** The robot needs to navigate to a chest of drawers. I will examine the video frames to identify the chest of drawers and select the frame index that shows the chest of drawers clearly. The chest of drawers appears in frames 71, 72, and 78. Frame 71 shows the chest of drawers clearly, so I will select that frame index.
**frame_indices:** [71]
Meta Analysis: Selects a frame too distant from the target despite a closer one being available

✗ **Text Agent:** The goal is to navigate to a chest of drawers. From the history, we see that the agent navigated through various rooms including the bedroom and living room. The dresser mentioned in the placing action at frame index 672 could be a chest of drawers. Therefore, the most relevant frame index to achieve the goal is 672.
**frame_indices:** [672]
Meta Analysis: Selects a frame where target objects is not visible but was visible in nearby frames

### Interaction

**Task:** Navigate to any receptacle you placed an object on.

✓ **GPT-4o:** To find a frame where the robot placed an object on a receptacle, I analyzed the video frames. In Frame 15, the robot places a lunchbox on a table. This frame clearly shows the object being placed, which matches the goal of navigating to a receptacle where an object was placed. Thus, Frame 15 is the most relevant frame for the task.
**frame_indices:** [15]

✗ **Gemini-2.0-flash:** Here's a breakdown of the objects picked and placed in the video, to help answer the user's question: * **Spray bottle:** Picked from a chair (00:03), placed on a nightstand (01:16). * **Lunchbox:** Picked from a chair (00:03), placed on a bathroom sink (00:25). * **Apple:** Picked from a table (00:07), kept on the ottoman (01:34). * **Shoe:** Picked from shelf (00:33), placed under lamp (00:41). * **Alarm clock:** Picked from a side table (00:55), kept on bathroom sink (01:05). * **Toy:** Picked from a table (01:27), placed on a chair (01:33). The user wants to navigate to any receptacle where an object was placed. So, we can choose any of the destination receptacles. Let's pick the ottoman (where the apple was placed). NUM_TARGETS_TO_REVISIT: ||1||. TIMESTAMP_INDEX: ||01:34||
**frame_indices:** [94]
Meta Analysis: Misidentifies the location of placing the apple which is the table

✗ **Qwen-2.5-VL:** The robot needs to navigate to a receptacle where it previously placed an object. From the video, we can see that the robot has placed various objects on different surfaces such as tables, chairs, and dressers. To achieve the goal, the robot should identify the last object it placed and then move to the receptacle where it was placed.
**frame_indices:** [12, 25, 47, 63, 78, 93]
Meta Analysis: Provides incorrect reasoning and identifies multiple frames (some incorrect) instead of single frame

✗ **Gemma-3:** The robot needs to navigate to the receptacles where it placed objects. Let's analyze the video to identify those frames. 1. **Frame 11:** The robot places a suitcase on a table. 2. **Frame 26:** The robot places a suitcase on a sink. 3. **Frame 65:** The robot places a clock on a sink. 4. **Frame 93:** The robot places a suitcase on a chair. Therefore, the frame indices to navigate to are 11, 26, 65, and 93.
**frame_indices:** [11, 26, 65, 93]
Meta Analysis: Incorrect event tracking and reasoning leads to misidentified frames

✓ **Text Agent:** The goal is to navigate to any receptacle where an object has been placed. From the history, the last action was placing an alarm clock on the dresser in the bedroom. Therefore, I should navigate to the bedroom.
**frame_indices:** [576]
Meta Analysis: Incorrect event tracking but still selects a valid frame

Figure 21: FINDINGDORY Spatial Tasks for the sample episode in Figure 19. Each VLM model uses 96 subsampled frames. *Text Agent* processes full video as defined in appendix D.4.

# Single-Goal Spatial Tasks

## Conditional Interaction

**Task:** Navigate to the receptacle that you picked the backpack from.

✗ **GPT-4o:** The task requires the robot to return to the receptacle where it previously picked up the backpack. By reviewing the sequence of frames in the video, frames 3 through 17 show the robot's interaction with the backpack. The backpack is first clearly seen on a chair in frame 3, and the robot interacts with it until frame 17. After frame 17, the robot seems to have moved away from the chair with the backpack. Thus, the frame closest to the object (the chair with the backpack) before it was picked is frame 3.
**frame_indices:** [3]
Meta Analysis: Ignores backpack that was picked from the table and focuses on "distractor" backpack that agent never interacted with

✗ **Gemini-2.0-flash:** Here's a breakdown of the video and the solution: * **Video Summary:** * The robot picks up a green spray bottle and a dinosaur lunchbox from a chair (00:03). * The robot picks up a red apple from a table (00:07). * The robot picks up the dinosaur lunchbox from a table (00:15). * The robot places the dinosaur lunchbox on a bathroom sink (00:26). * The robot picks up a black-and-blue sneakers from a shelf (00:33). * The robot places a black-and-blue sneakers on the cabinet next to a table lamp (00:41). * The robot picks up the green alarm clock (00:56). * The robot placed the alarm clock next to the lunch box on the bathroom counter (01:05). * The robot picks up the green spray bottle and panda toy from the cabinet next to the lamp (01:16). * The robot places the green spray bottle and the lunchbox back on the chair (01:33). * **Task:** Navigate to the receptacle that you picked the backpack from. * **Solution:** The robot picked the backpack from the chair at 00:03. Therefore, the response is: NUM_TARGETS_TO_REVISIT: ||1||. TIMESTAMP_INDEX: ||00:03||
**frame_indices:** [3]
Meta Analysis: Correctly tracks events but makes error in reasoning over the events

✓ **Qwen-2.5-VL:** The robot needs to navigate back to the location where it picked up the backpack. From the video, the backpack was picked up on frame 16 when the robot was in the bathroom. Therefore, the robot should move to the frame where the backpack was last seen.
**frame_indices:** [16]
Meta Analysis: Backpack was dropped in the bathroom

✓ **Gemma-3:** The robot picked up the backpack from a receptacle. I need to find the frame index where the receptacle is visible. The backpack was picked up in frames 15-16. The receptacle is the small table with a dinosaur-themed bag on it. The frame index where the receptacle is visible is frame 15.
**frame_indices:** [15]

✗ **Text Agent:** The goal is to navigate to the receptacle where the backpack was picked. From the history, we see that the backpack was picked in the bedroom at frame index 384. Therefore, we should navigate to the bedroom.
**frame_indices:** [384]
Meta Analysis: Incorrect event tracking

## Spatial Relationship

**Task:** Navigate to the receptacle that you picked an object from which is the farthest from your current location.

✗ **GPT-4o:** The robot picked up objects from various receptacles during the navigation. To determine the farthest receptacle from the current location, it's important to identify the locations where the robot previously picked up items and then calculate which of these locations is the farthest from the end point of the video. Key frames to consider include: - Frame 4: Picked up an item from the chair near the glass door. - Frame 17: Picked up an item from a table in what seems to be the living area. - Frame 41: Picked up a shoe from a different location. The current location (final frame) seems to indicate the robot is in a bedroom. Relative to this, the farthest picking location would likely be Frame 4, where the robot picked an item up from a chair near the glass door. This probable distance is because the living room and shoe area seem to be closer to the bedroom than the chair near the glass door. Thus, the robot should navigate to the receptacle at Frame 4.
**frame_indices:** [4]
Meta Analysis: Incorrect description of final agent location as bedroom. In addition, chair from which item was picked is visible clearly visible in final frame

✗ **Gemini-2.0-flash:** Here's how we can solve this problem: 1. **Summarize the Robot's Actions:** * **Chair:** * Picked up a spray bottle and a lunchbox from the chair near the sliding doors (00:03). * Placed the spray bottle on the nightstand next to the lamp in the bedroom (01:16). * Picked up an apple from the table in the balcony (00:07). * Dropped the lunchbox on the bathroom sink (00:26). * **Table** * Picked up a shoe from the shelf behind the sofa (00:33). * Dropped the shoe to the nightstand (00:41). * **Washing Machine** * Picked up an alarm clock from the top of washing machine(00:56). * Dropped the alarm clock near the lunchbox in the bathroom sink (01:08). * **Chair near sofa** * Picked up a toy from the chair near the sofa(01:27). * Dropped the lunchbox near the chair in the balcony (01:35). 2. **Determine Distances:** * The chair near the sliding doors and washing machine appears to be the farthest receptacles from current position (frame 00:00). 3. **Identify the First Receptacle:** The first receptacle is the chair near the sliding doors. 4. **Final Answer:** Based on the analysis, the final answer is: NUM_TARGETS_TO_REVISIT: ||1||. TIMESTAMP_INDEX: ||00:03||
**frame_indices:** [3]
Meta Analysis: Incorrect event tracking and no clear spatial reasoning exhibited

✗ **Qwen-2.5-VL:** The robot needs to navigate to the farthest receptacle from its current location. It has previously picked up a red object from a table and a lunchbox from a sink. The red object is on a table, and the lunchbox is on a sink. The table is located in the dining area, and the sink is in the bathroom. The dining area is further away from the current location than the bathroom. Therefore, the robot should move towards the dining area.
**frame_indices:** [12]
Meta Analysis: Agent final location is close to the chair from which the red object was picked

✗ **Gemma-3:** The robot's goal is to navigate to the receptacle that it picked an object from which is the farthest from its current location. The robot picked up a blue object from a table in the first few frames. The receptacle is the table. The table is located in the outdoor area of the house. The robot is currently in the bathroom. Therefore, the robot should navigate to frame 84, which shows the table in the outdoor area.
**frame_indices:** [84]
Meta Analysis: Incorrect event tracking and spatial reasoning

✗ **Text Agent:** To determine the farthest receptacle from the current location, we need to calculate the distance between the agent and each receptacle it has interacted with. However, since we don't have explicit distance data, we'll assume the order of rooms visited reflects the order of interaction. The farthest room from the starting point would likely be the last one visited. In this case, the last room visited is the 'living room' at frame index 1308.
**frame_indices:** [1308]
Meta Analysis: No spatial reasoning and reliance on unfounded assumptions

Figure 22: FINDINGDORY Spatial Tasks for the sample episode in Figure 19. Each VLM model uses 96 subsampled frames. *Text Agent* processes full video as defined in appendix D.4.

## Single-Goal Spatial Tasks

### Object Attributes

**Task:** Navigate to a red colored object again that you interacted with yesterday.

✓ **GPT-4o:** The video shows consecutive frames captured by a robot inside a house. The task is to identify the frame where the robot had an interaction with a red-colored object. Upon reviewing the frames, the robot interacted closely with a red apple. This interaction can be observed between frames 6 and 11, with the robot's gripper appearing close to the apple. Therefore, the frame where the robot seems closest to the apple is frame 7.
**frame_indices:** [7]

✓ **Gemini-2.0-flash:** The robot picks up the following objects and places them at different locations: - A green spray bottle and a lunch box from a chair and places them on a table in the bathroom (frame 26). - A red apple from a table and does not drop it, implies it is still holding it (frame 11). - A shoe from a shelf and drops it next to a lamp in a bedroom (frame 41). - An alarm clock from a counter and drops it near the lunch box in the bathroom (frame 65). - A panda toy from a shelf in a bedroom and drops it near the spray bottle (frame 77). The user wants to revisit the red colored object (red apple). Since the robot never dropped it, we should revisit the table where it picked up the apple to see if it placed it back there. NUM_TARGETS_TO_REVISIT: ||1||. TIMESTAMP_INDEX: ||00:07||
**frame_indices:** [7]
**Meta Analysis:** Does not identify the dropping of apple on table but selects correct frame based on where the apple was viewed

✓ **Qwen-2.5-VL:** The robot needs to navigate back to the red object it interacted with yesterday. From the video frames, the red object appears on a table in the living room at frame 10.
**frame_indices:** [10]

✗ **Gemma-3:** The robot's goal is to navigate to a red colored object that it interacted with yesterday. Looking through the video, I can see the robot interacting with a red suitcase multiple times. The first time the robot interacts with the red suitcase is around frame 15. The robot then interacts with the suitcase again around frame 25, 66, 68, 86, 92, and 95. I will choose frame 15 as the frame index to navigate to the red suitcase.
**frame_indices:** [15]
**Meta Analysis:** Fails to identify the red apple object that was dropped on the table

✗ **Text Agent:** To accomplish the goal of navigating to a red colored object, I need to identify the red object from the history. From the given history, there is no explicit mention of a red object. However, if we assume the red object was picked or placed at some point, we should look for the most recent navigation action involving a red object. Since the history does not specify the color, we'll choose the last navigation action as it might be the most recent interaction.
**frame_indices:** [1308]
**Meta Analysis:** Fails to identify the red apple object that was dropped on the table

### Room Visitation

**Task:** Navigate to the room where you placed the backpack in.

✗ **GPT-4o:** The robot needs to find the room where it placed the backpack. While reviewing the frames, the backpack appears prominently in frames 3, 25, and 93, among others. The most recent placement seems to be in frame 93, which occurs at time 11:06, showing the backpack is located on a chair initially visible in the earlier part of the video. This suggests that the robot placed the backpack there later in the sequence since it was previously interacting with it near the bathroom sink (frame 25). Thus, the correct location the robot should navigate to is found in frame 93.
**frame_indices:** [93]
**Meta Analysis:** Focuses on "distractor" backpack that agent never interacted with

✓ **Gemini-2.0-flash:** Here's a breakdown of the objects picked and placed, to help answer the question: * **Spray Bottle:** Picked from the chair at 00:03, placed on the nightstand at 01:16 * **Red Apple:** Picked from the table at 00:07, placed on the chair at 00:03 * **Shoe:** Picked from shelf at 00:33, placed near the lamp at 00:41 * **Lunchbox:** Picked from table at 00:15, placed on bathroom counter at 00:26 * **Clock:** Picked from the drawer at 00:56, placed on the lunchbox on bathroom counter at 01:05 * **Panda Toy:** Placed on the nightstand at 01:16, picked from the nightstand lamp table at 01:16, placed on table at 00:15 The lunchbox (referred to as "backpack" in the prompt) was placed on the bathroom counter. So, to navigate to where the backpack was placed, we should go to the bathroom. NUM_TARGETS_TO_REVISIT: ||1||. TIMESTAMP_INDEX: ||00:26||
**frame_indices:** [26]

✓ **Qwen-2.5-VL:** The robot needs to navigate to the room where it placed the backpack. From the video, we can see the backpack being placed on a sink in a bathroom. The robot should move to this location to find the backpack.
**frame_indices:** [25]

✗ **Gemma-3:** The robot needs to navigate to the room where the backpack was placed. From the video, the backpack was placed on a chair in a room with a table and a plant. The frames showing the backpack on the chair are frames 93 and 95. I will choose frame 93 as it is a clearer view of the backpack and the surrounding environment.
**frame_indices:** [93]
**Meta Analysis:** Focuses on "distractor" backpack that agent never interacted with

✗ **Text Agent:** To determine the room where the backpack was placed, I need to find the frame where the agent was placing an object. From the history, the last frame where the agent was placing an object was at frame 672, where the alarm clock was placed on the dresser in the bedroom. Since the goal is to navigate to the room where the backpack was placed, we should look for a frame where the agent was navigating to a specific room. The next navigation frame is at 768, which is in the bathroom. However, there is no indication that the backpack was placed in the bathroom. The next navigation frame is at 864, which is in the bathroom as well. The next navigation frame is at 960, which is in the bedroom. Therefore, the backpack was likely placed in the bedroom.
**frame_indices:** [960]
**Meta Analysis:** Misses tracking frames where backpack is picked from table in living room and placed in bathroom sink

Figure 23: FINDINGDORY Spatial Tasks for the sample episode in Figure 19. Each VLM model uses 96 subsampled frames. *Text Agent* processes full video as defined in appendix D.4.

## Single-Goal Temporal Tasks

### Interaction Order

**Task:** Navigate to the object you that you interacted with between the interactions with apple and shoe.

✓ **GPT-4o:** First, identify the interactions involving the apple and shoe. The apple interaction occurs at frames 7 to 11. The shoe interaction occurs at frames 33 to 43. Between these interactions, the robot interacted with a lunchbox. The relevant frames for this interaction are from 25 to 28 and 64 to 68. The robot should navigate to the lunchbox in this task.
**frame_indices: [26]**

✗ **Gemini-2.0-flash:** Here's a breakdown of the robot's actions in the video: 1. **00:03**: Robot picks up a spray bottle and lunchbox from a chair. 2. **00:07**: Robot picks up an apple from a table. 3. **00:15**: Robot places the lunchbox on another table. 4. **00:25**: Robot places the lunchbox on the bathroom sink. 5. **00:33**: Robot picks up a shoe from a shelf. 6. **00:41**: Robot places the shoe under a lamp on a nightstand. 7. **00:55**: Robot picks up an alarm clock from a table. 8. **01:05**: Robot places the alarm clock near the lunchbox on the bathroom sink. 9. **01:16**: Robot places the spray bottle on a nightstand. The object interacted with between the apple and shoe is the lunchbox. It was picked up at 00:03 and placed on the sink at 00:25. NUM_TARGETS_TO_REVISIT: ||1||. TIMESTAMP_INDEX: ||00:03||
**frame_indices: [3]**
Meta Analysis: Correct event tracking and target identification but incorrectly navigates to the object picking location instead of final placement location

✗ **Qwen-2.5-VL:** The robot needs to navigate from the apple interaction to the shoe interaction. The apple interaction occurred at frame 10, and the shoe interaction occurred at frame 35. Therefore, the robot should move to the frame right after the apple interaction.
**frame_indices: [11]**
Meta Analysis: Fails to identify the backpack target object for navigation

✗ **Gemma-3:** The robot interacted with an apple in frame 7 and a shoe in frame 43. The object between these interactions is the green alarm clock. The robot interacted with the alarm clock in frame 55. Therefore, the frame index to move to is 55.
**frame_indices: [55]**
Meta Analysis: Incorrect target object identification due to temporal reasoning error

✗ **Text Agent:** The goal is to navigate to the object you interacted with between the interactions with apple and shoe. From the history, the interaction with the apple occurred at frame 672 and the interaction with the shoe occurred at frame 384. Therefore, we need to find the frame index between these two events.
**frame_indices: [576]**
**Meta Analysis: Incorrect event tracking and ignores the initial interaction with the apple**

### Time-Based

**Task:** Navigate to the object that you interacted with at 8:26 yesterday.
**For this task, the query timestamp can be slightly different but target solution object remains constant

✓ **GPT-4o:** To find the frame index where the robot should move, we need to first identify the object that was interacted with at 8:06. Reviewing the timestamps on the images, we can see that the robot interacted with an object around that time. At 8:08 (frame 39), the robot is near the nightstand with a lamp and small weights. The robot's interaction here suggests it is the object of interest. Moving a bit further into frames, the shoe is being interacted with from frame 39 to frame 41. Therefore, the frame closest to this moment, and thus relevant for the task, is frame 39.
**frame_indices: [39]**

✓ **Gemini-2.0-flash:** The robot performs the following actions: - Picks up a spray bottle from the chair (00:03), - Picks up an apple from the table (00:07), - Places lunchbox on bathroom sink (00:26), - Picks up a shoe from the shelf (00:33), - Places shoe on the nightstand (00:41), - Picks up an alarm clock from the table next to the washing machine (00:56), - Places alarm clock on the bathroom sink next to lunchbox (01:05), - Picks up a panda figurine next to the lamp (01:16), - Places spray bottle next to the lamp (01:17). The user is asking to revisit the object that was interacted with at 08:21, which is the shoe. NUM_TARGETS_TO_REVISIT: ||1||. TIMESTAMP_INDEX: ||00:41||
**frame_indices: [41]**

✓ **Qwen-2.5-VL:** The robot needs to navigate back to the object it interacted with at 8:26 yesterday. From the video frames, the object it interacted with appears to be a shoe on the floor near the bed at frame 43.
**frame_indices: [43]**

✗ **Gemma-3:** The robot interacted with a suitcase at 08:27. I need to find the frame index where the robot is closest to the suitcase. Looking at the images, the suitcase is visible in frames 64, 65, 66, 67, 68, 69, 88, 90, 92, 93, 94, 95. Frame 93 seems to be the closest to the suitcase.
**frame_indices: [93]**
Meta Analysis: Identifies the incorrect target object. In addition, focuses on the "distractor" backpack which was not interacted with

✗ **Text Agent:** To determine the frame index corresponding to the interaction at 8:27 yesterday, I need to find the timestamp closest to 8:27. The timestamps provided are in a 24-hour format, so 8:27 would be 20:27 in military time. The closest timestamp in the history is 09:27, which corresponds to the frame indices 864 and 960. Since the goal is to navigate to the object interacted with at 8:27, we should choose the frame index that aligns with this timestamp.
**frame_indices: [864]**
Meta Analysis: Erroneous conversion of query timestamp to a new value (09:27). But retrieves the correct object corresponding to interaction at 09:27

Figure 24: FINDINGDORY Temporal Tasks for the sample episode in Figure 19. Each VLM model uses 96 subsampled frames. *Text Agent* processes full video as defined in appendix D.4.

**Single-Goal Temporal Tasks**

**Duration Tracking**

**Task:** Navigate to to the room that you spent most time in.

✓ **GPT-4o:** I need to identify which room the robot spent the most time in during the navigation. Analyzing the frames, the setting shifts around a few distinct locations, primarily different rooms in the house. The living room appears frequently, with repetitive views of chairs, tables, and decorative setups. The frames in the living room span from the start, go through frames with consistent furniture and decor like in frames 1-12, and reappear frequently towards the latter part, such as around frames 50-60ish. Other parts like the bathroom or kitchen have fewer frames indicating less time spent. Thus, identifying the living room as the room where the robot spent the most time, the earliest close frame in that setting is a good reference for moving to the dominant room setting.
**frame_indices:** [1]

✗ **Gemini-2.0-flash:** The robot spends most of its time in the living room and the bedroom. Since both rooms are spent similar amount of time, I will default to choosing the living room. NUM_TARGETS_TO_REVISIT: ||1||. TIMESTAMP_INDEX: ||01:34||
**frame_indices:** [94]
Meta Analysis: Decides to choose the living room but selects a frame where agent is in the smaller patio area

✗ **Qwen-2.5-VL:** The robot needs to navigate to the room it spent the most time in. Based on the frames, the robot spends more time in the bedroom, as indicated by the higher frame numbers and the presence of personal items like a bed, nightstand, and alarm clock. Therefore, the robot should move to the bedroom.
**frame_indices:** [40]
Meta Analysis: Ignores the large proportion of frames that are situated in the living room

✗ **Gemma-3:** The robot spent most of its time in the room with the blue floor. The video starts in a room with a blue wall, then moves to a room with a white floor, then to a room with a blue floor. The robot spends a significant amount of time in the room with the blue floor, moving around and interacting with various objects. The robot then moves to a room with a white floor, then to a room with a blue wall, then to a bathroom, and then back to the room with the blue floor. Therefore, the robot should navigate to the room with the blue floor.
**frame_indices:** [1, 51, 52, 58, 60, 81, 82, 83, 92, 93, 94, 95]
Meta Analysis: Does not reason about the general semantic structure of the rooms. Selects multiple correct frames based on matching the floor color but selects incorrect frames from the patio area due to similar floor color

✓ **Text Agent:** To determine the room where the agent spent the most time, I need to calculate the duration of time spent in each room. The timestamps provide the start and end times for each navigation event. By subtracting the start timestamp from the end timestamp, we can calculate the duration of time spent in each room. The room with the longest duration is the one where the agent spent the most time.
**frame_indices:** [1248]
Meta Analysis: Selects correct frame but does not provide concrete duration spent in each room

Figure 25: FINDINGDORY Temporal Tasks for the sample episode in Figure 19. Each VLM model uses 96 subsampled frames. *Text Agent* processes full video as defined in appendix D.4.

## Multi-Goal Tasks

### Unordered Revisitation

**Task:** Revisit all the receptacles you picked objects from yesterday.

✓ **GPT-4o:** The video shows the robot navigating through a house while picking up objects from various receptacles. To complete the task, the robot needs to revisit the same receptacles where objects were picked up. 1. Frame 3 shows the robot picking an object from the chair. Revisit this chair. 2. Frame 7 shows the robot picking an object (an apple) from the dining table. Revisit this table. 3. Frame 15 shows the robot picking an object from a round table beside a chair. Revisit this table. 4. Frame 33 shows the robot picking a shoe from a low shelf. Revisit this shelf. 5. Frame 55 shows the robot picking a clock from a countertop. Revisit this countertop. 6. Frame 76 shows the robot picking a spray bottle from a side table near a lamp. Revisit this table. These are the main locations the robot picked objects from, so these frames need to be revisited.
frame_indices: [3, 15, 33, 55, 76]
Meta Analysis: Incorrectly identifies apple being picked from table whereas the apple was dropped on table. But omits index 7 in final solution without any clear reason

✗ **Gemini-2.0-flash:** Here's a breakdown of the objects picked and placed, to determine the receptacles that need revisiting: * **Spray bottle & Lunchbox:** * Picked from: Chair in the living room (00:03) * Placed on: Chair in the living room (00:04), then plant table (00:15), then sink (00:26). * **Apple:** * Picked from: Plant table (00:07) * Placed on: Plant table (00:11). * **Shoe:** * Picked from: Shelf in living room (00:33) * Placed on: Nightstand in bedroom (00:41). * **Lunchbox:** * Picked from: Sink in bathroom (00:27) * Placed on: Ottoman (01:24) * **Clock:** * Picked from: Washing Machine (00:56) * Placed on: Sink in bathroom (01:08) * **Panda Toy:** * Picked from: Nightstand in bedroom (01:16) * Placed on: Chair in living room (01:27) Now, let's identify the timestamps for revisiting the original receptacles: 1. **Chair in living room:** Where the spray bottle and lunchbox were initially placed. (00:03) 2. **Plant table:** Where the apple was initially placed. (00:07) 3. **Shelf in living room:** Where the shoe was initially placed. (00:33) 4. **Washing machine:** Where the clock was initially placed. (00:56) 5. **Nightstand in bedroom:** Where the panda toy was initially placed. (01:16) 6. **Sink in bathroom** Where the lunchbox was initially placed. (00:27) NUM_TARGETS_TO_REVISIT: ||6||.
TIMESTAMP_INDEX: ||00:03,00:07,00:33,00:56,01:16,00:27||
frame_indices: [3, 7, 33, 56, 76, 27]
Meta Analysis: Incorrect event tracking and misidentifies backpack placement frames as pick frames

✗ **Qwen-2.5-VL:** The robot needs to revisit all the receptacles it picked objects from yesterday. From the video, we can see that the robot picked up objects from various receptacles such as a table, a suitcase, a nightstand, and a dresser. To achieve the goal, the robot should revisit these locations in the order they were visited.
frame_indices: [12, 16, 34, 50]
Meta Analysis: Inadequate reasoning and hallucinates suitcase as a receptacle

✗ **Gemma-3:** Error parsing output
frame_indices: [-1]
Meta Analysis: Model fails to produce any coherent output

✗ **Text Agent:** To revisit all the receptacles from which objects were picked yesterday, I need to identify the rooms where objects were picked and then navigate back to those rooms. From the history, we see that objects were picked from the living room, kitchen, bedroom, and bathroom. Therefore, I should navigate to these rooms.
frame_indices: [0, 96, 192, 288, 384, 480, 576, 672, 768, 864, 960, 1056, 1152, 1248]
Meta Analysis: Incorrectly focuses only on room revisitation instead of revisiting specific receptacles

### Ordered Revisitation

**Task:** Revisit all the receptacles you placed objects on yesterday in specific order. The order to revisit them is: fifth, second, first, third.

✗ **GPT-4o:** The robot needs to revisit specific receptacles in order. To do this, it needs to identify the frames where objects were placed: 1. **First Receptacle (Fifth in order of placement):** - Frame 64: Lunchbox on the sink in the bathroom. 2. **Second Receptacle (Second in order of placement):** - Frame 7: Apple on the table in the dining area. 3. **Third Receptacle (First in order of placement):** - Frame 16: Lunchbox on the small table in the living room. 4. **Fourth Receptacle (Third in order of placement):** - Frame 33: Shoe on the shelf in the entryway. The robot should start with the lunchbox on the sink (fifth placement), then the apple on the table (second placement), revisit the lunchbox in the living room (first placement), and finally go to the shoe on the shelf (third placement).
frame_indices: [64, 7, 16, 33]
Meta Analysis: Unable to list out the original object placement events and then adapt to the query revisitation sequence

✗ **Gemini-2.0-flash:** Here's a summary of the objects picked and placed, and their locations: * **Apple:** Picked from the table near the plant (00:07), placed on the table on the balcony (00:03). * **Spray bottle:** Picked from the chair on the balcony (00:03), placed under the lamp on the nightstand (01:16). * **Shoe:** Picked from the shelf (00:33), placed under the lamp on the nightstand (01:41). * **Lunchbox:** Picked from the table (00:15), placed on the bathroom counter (00:25). * **Clock:** Picked from the table next to the washing machine (00:55), placed on the bathroom counter next to lunchbox (01:05). * **Panda:** Picked from under the lamp on the nightstand, placed on the side table in living room (01:27). Given the objects and placements, let's identify the receptacles for revisitation in the required order: 1. The fifth item that was placed was the **clock** on the bathroom counter. 2. The second item that was placed was the **spray bottle** under the lamp on the nightstand. 3. The first item that was placed was the **apple** on the chair on the balcony. 4. The third item that was placed was the **shoe** under the lamp on the nightstand. Here are the timestamps: * **Lunchbox and clock on the bathroom counter:** 00:25 * **Spray bottle and shoe on the nightstand:** 01:16 * **Apple on the chair on the balcony:** 00:03 ```text NUM_TARGETS_TO_REVISIT: ||4||. TIMESTAMP_INDEX: ||00:25,01:16,00:03,01:16|| ```
frame_indices: [25, 76, 3, 76]
Meta Analysis: Correctly lists out original placement sequence although hallucinates interaction with spray bottle. Confuses picking and placement timestamps while generating final solution

✗ **Qwen-2.5-VL:** The robot needs to revisit the receptacles it placed objects on yesterday in the specified order: fifth, second, first, third. To achieve this, we need to identify the frames where these receptacles were visited. The frames provided show the robot interacting with various objects and moving through different rooms. We need to find the frames corresponding to the receptacles in the given order.
frame_indices: [59, 13, 26, 48]
Meta Analysis: **No clear reasoning on how the original placement sequence was adapted to the query revisitation seqeuence**

✗ **Gemma-3:** Error parsing output
frame_indices: [-1]
Meta Analysis: Model fails to produce any coherent output

✗ **Text Agent:** To accomplish the goal of revisiting the receptacles in the specified order, I need to identify the receptacles that were placed with objects. From the history, the receptacles mentioned are 'dresser' and 'sink'. The order to revisit them is fifth, second, first, third. Based on the timestamps and room names, the 'dresser' was visited at frame indices 672 and 768, and the 'sink' was visited at frame indices 288 and 384. The order given is fifth, second, first, third, so we should start with the 'dresser' (frame indices 672 and 768), then the 'sink' (frame indices 288 and 384).
frame_indices: [672, 768, 288, 384]
Meta Analysis: Only lists out two receptacles on which objects were placed but task requires revisiting all of them

Figure 26: FINDINGDORY Multi-Goal Tasks for the sample episode in Figure 19. Each VLM model uses 96 subsampled frames. *Text Agent* processes full video as defined in appendix D.4.

Table 5: Task Categories with all associated prompts

| Memory Type | Example Instructions |
|---|---|
| Object Recall | Navigate to a {target_object_name}. |
| | Navigate to a {target_receptacle_name}. |
| Interaction | Navigate to any receptacle you interacted with. |
| | Navigate to any receptacle you did not interact with. |
| | Navigate to any object that you interacted with yesterday. |
| | Navigate to any object that you did not interact with yesterday. |
| | Navigate to any receptacle you picked an object from. |
| | Navigate to any receptacle you placed an object from. |
| Conditional Interaction | Navigate to the {target_object_name} that you interacted with yesterday. |
| | Navigate to a {target_object_name} that you did not interact with yesterday. |
| | Navigate to a {target_receptacle_name} you did not interact with yesterday. |
| | Navigate to a {target_receptacle_name} you picked an object from. |
| | Navigate to a {target_receptacle_name} you placed an object on. |
| | Navigate to the receptacle that you picked the {target_object_name} from. |
| | Navigate to the object that you picked from the {target_receptacle_name}. |
| Object Attributes | Navigate back to a {target_shape} shaped object that you interacted with yesterday. |
| | Navigate back to a {target_color} colored object that you interacted with yesterday. |
| | Navigate to an interacted object with {target_print_or_design} on it. |
| | Find an already interacted object that is made of {target_material}. |
| | Go back to an interacted object that is used for {target_functionality}. |
| Spatial Relationship | Navigate to the receptacle that you interacted with which is the farthest from your current location. |
| | Navigate to the receptacle that you did not interact with which is the farthest from your current location. |
| | Navigate to the receptacle that you picked an object from which is the farthest from your current location. |
| | Navigate to the receptacle that you placed an object on which is the farthest from your current location. |
| | Navigate to the object which you interacted with which is the farthest from your current location. |
| Room Visitation | Navigate to the room where you picked the {target_appearance_order} object from. |
| | Navigate to the room where you placed the {target_appearance_order} object in. |
| | Navigate to the room where you picked the {target_object_name} from. |
| | Navigate to the room where you placed the {target_object_name} in. |
| | Navigate to a room that you did not visit yesterday. |
| | Navigate to the {target_appearance_order} object that you interacted with yesterday. |
| | Navigate to the {target_appearance_order} receptacle that you picked an object from. |
| | Navigate to the {target_appearance_order} receptacle that you placed an object on. |
| | Navigate to the receptacle that you picked the {target_appearance_order} object from. |
| | Navigate to the object that you picked from the {target_appearance_order} receptacle. |
| | Navigate to the object you interacted with immediately after ending the interaction with {target_object_name}. |
| | Navigate to the object you interacted with immediately before interacting with {target_object_name}. |
| | Navigate to the object you interacted with {target_num_counts} interactions after {target_object_name}. |
| | Navigate to the object you interacted with {target_num_counts} interactions before {target_object_name}. |
| | Navigate to the object that you interacted with between the interactions with {target_object_name_1} and {target_object_name_2}. |
| | Navigate to the receptacle that you placed an object on right before you started interacting with {target_object_name}. |
| | Navigate to the receptacle that you picked an object from right after you finished interacting with {target_object_name}. |

*Continued on the next page...*

**Interaction Order**

| Memory Type | Example Instructions |
|---|---|
| | Navigate to the receptacle that you placed an object on {target_num_counts} interactions before you started interacting with {target_object_name}. |
| | Navigate to the receptacle that you picked an object from {target_num_counts} interactions after you finished interacting with {target_object_name}. |
| | Navigate to the receptacle that you placed an object on between the interactions with {target_object_name_1} and {target_object_name_2}. |
| | Navigate to the receptacle that you picked an object from between the interactions with {target_object_name_1} and {target_object_name_2}. |
| **Time-Based** | Navigate to the receptacle that you interacted with at {XX:XX} yesterday. |
| | Navigate to the object that you interacted with at {XX:XX} yesterday. |
| **Duration Tracking** | Navigate to the object which took the longest time to rearrange. |
| | Navigate to the room that you spent the most time in. |
| | Navigate to the object which took the shortest time to rearrange. |
| **Unordered Revisitation** | Revisit all the receptacles you picked objects from yesterday. |
| | Revisit all the receptacles you placed objects on yesterday. |
| | Revisit all the {target_receptacle_name} you placed objects on yesterday. |
| | Revisit all the {target_receptacle_name} you picked objects from yesterday. |
| | Revisit all the objects you interacted with yesterday. |
| | Revisit all the receptacles you interacted with yesterday. |
| **Ordered Revisitation** | Revisit all the receptacles you picked objects from yesterday in specific order. |
| | Revisit all the receptacles you placed objects on yesterday in specific order. |
| | Revisit all the objects you interacted with yesterday in specific order. |

