# OpenReview forum: "FindingDory: A Benchmark to Evaluate Memory in Embodied Agents"
_ICLR.cc/2026/Conference — Submitted to ICLR 2026_

### Official Review · Reviewer_63kF · 2025-10-26

**Soundness:** 2
**Presentation:** 2
**Contribution:** 3
**Rating:** 6
**Confidence:** 4

**Summary:**

The authors introduce FindingDory, a new benchmark designed to evaluate long-term memory in embodied agents, addressing the limitations of VLMs in handling long-term experiences. Set in the Habitat simulator, this benchmark includes 60 procedurally extensible tasks that require agents to reason over vast image collections from past interactions to solve spatial, temporal, and semantic challenges. The paper evaluates several state-of-the-art VLMs, including GPT-40 and Gemini-2.0-Flash, and finds they struggle significantly, with the best proprietary models achieving success rates below 28%. These models performed especially poorly on multi-goal tasks and temporal reasoning, and their performance often degraded when provided with more video frames, highlighting the need for more efficient memory mechanisms.

**Strengths:**

+ The paper addresses a crucial and underexplored area in embodied AI: evaluating long-horizon memory in realistic, interactive scenarios. The introduction persuasively argues for the gap between static video QA benchmarks and the memory requirements of active embodied tasks.
+ The benchmark uses well-specified, multi-faceted metrics, such as HL-SR/HL-SPL for high-level goal selection and LL-SR/LL-SPL for navigation. It also includes relaxed variants like DTG-SR and SC-SR to diagnose specific failure modes. The success criteria are concrete, defined by distance, angle, semantic coverage, and room region, and the evaluation procedures are clearly detailed.
+ A broad, templated task suite is provided, featuring 60 templates that cover spatial, temporal, and semantic reasoning and include multi-goal variations. This task suite is also procedurally extensible, allowing for scalable evaluation.
+ The paper delivers concrete, category-level findings. For example, it shows that VLMs can semantically detect large receptacles but fail to precisely localize them (where SC-SR is much higher than DTG-SR). Other findings include a significant gap between HL-SR and HL-SPL, which highlights VLM failure in selecting the nearest correct entity , and the extreme difficulty models face with multi-goal tasks.

**Weaknesses:**

+ The evaluation treats "unsolvable" instructions as failures. Although the paper quantifies that 3.85% of tasks are "Originally Unsolvable", the agent is not given a way to abstain or declare a task unsolvable and must output frame indices, which diverges from realistic agent requirements.
+ Despite being mentioned by the authors as one of the limitations, data-generation artifacts may confound memory signals; specifically, the "magic grasp"  used by the oracle agent for pick-and-place actions creates abrupt, non-naturalistic transitions. This could weaken temporal cues and confuse VLMs trying to track interactions.
+ Attribute tasks rely on GPT-4o-generated object descriptions and are validation-only. This reduces reproducibility/consistency between train and val and may introduce bias from an external model.
+ A navigation bottleneck in the hierarchical policy evaluation may mask the VLM's true memory quality. The overall success rate (LL-SR) drops sharply when the high-level Qwen VLM is paired with a low-level ImageNav policy. This is attributed to a distribution shift, as VLM-selected goal frames (e.g., from pick-place routines) often provide poor visual cues compared to those the navigation policy was trained on. This makes it difficult to isolate the VLM's memory contribution in the end-to-end system.
+ The paper reports all key results as point estimates (e.g., Figure 2) without confidence intervals or standard deviations from multiple runs. While the high cost of full evaluations is acknowledged, this lack of uncertainty reporting also extends to smaller-scale analyses like the video length subsampling study and the cross-benchmark transfer experiments, where multi-run statistics might have been more feasible.

**Questions:**

1. How might the "unsolvable" task design and "magic grasp" data artifacts skew the evaluation of an agent's true memory capabilities?
2. Given the significant navigation bottleneck, how could the benchmark better isolate the VLM's high-level memory reasoning from low-level execution failures?
3. Considering the lack of uncertainty reporting and the use of external models for attribute tasks, what steps could improve the benchmark's statistical reliability and reproducibility?
4. Do authors plan to include diverse embodiments beyond the Fetch robot?

---

> ### Author Response · Authors · 2025-11-24
> **Rebuttal**
>
> ### Unsolvable Instructions Clarification
> > The evaluation treats "unsolvable" instructions as failures. Although the paper quantifies that 3.85% of tasks are "Originally Unsolvable", the agent is not given a way to abstain or declare a task unsolvable and must output frame indices, which diverges from realistic agent requirements.
>
>
> We would like to clarify what we mean by “unsolvable.” Our baselines adopt the dominant approach in current memory-augmented navigation systems (e.g. Mobility VLA), where the high-level module must select one or more frames from the recorded history as goal locations for the low-level controller. Under this formulation, some otherwise valid instructions become infeasible. For example, consider the task “Find an apple that you did not interact with yesterday.” If the agent never closely approached that apple during experience collection, then no frame in the history can serve as a goal frame as per our metrics. Since the baseline must output frame indices to specify goals, it cannot execute this task. This reflects a limitation of the baseline’s goal-parameterization rather than a limitation of the benchmark itself.
>
> Future research may develop end-to-end agents, memory modules, or goal representations that do not rely on frame selection and can therefore solve these tasks. We will revise the paper to explicitly clarify what we mean by “unsolvable” and to clarify that the benchmark does not restrict agents from declaring tasks unreachable or representing goals more expressively.
>
>
> ---------
> ### Data Generation Artifacts Clarification
> > Despite being mentioned by the authors as one of the limitations, data-generation artifacts may confound memory signals; specifically, the "magic grasp" used by the oracle agent for pick-and-place actions creates abrupt, non-naturalistic transitions. This could weaken temporal cues and confuse VLMs trying to track interactions.
>
>
> Unfortunately, this behavior is a limitation of the simulator. To mitigate this, as noted in line 818, we try to make the interaction as realistic as possible by first orienting the robot toward the object, visually centering it, and then moving its arm very close before the magic grasp action is executed. We also provide a link to our website in line 477, where the reviewer can see the pick and place maneuvers. This introduces a short but meaningful interaction period that should be sufficient for VLMs to interpret the event as a pick action rather than an instantaneous state change.
>
> ---------
> ### Attribute-based Tasks Annotations
> > Attribute tasks rely on GPT-4o-generated object descriptions and are validation-only. This reduces reproducibility/consistency between train and val and may introduce bias from an external model.
>
> Attribute tasks appear only in validation because the train and val splits use **disjoint object sets**, and generating descriptions for the training set would require annotating thousands of objects with descriptor attributes. For the validation split, we manually reviewed each and every GPT-4o generated attribute description to ensure accuracy and eliminate model-induced biases.  These instructions also function as **held-out task types**, ensuring that models do not overfit to the specific memory skills exercised during training and must generalize to a different phrasing of the tasks which many current approaches would fail at (ones that construct explicit semantic maps for example).

---

> ### Author Response · Authors · 2025-11-24
> **Rebuttal**
>
> ### High-Level/Low-Level Metrics Clarification
> > A navigation bottleneck in the hierarchical policy evaluation may mask the VLM's true memory quality. The overall success rate (LL-SR) drops sharply when the high-level Qwen VLM is paired with a low-level ImageNav policy. This is attributed to a distribution shift, as VLM-selected goal frames (e.g., from pick-place routines) often provide poor visual cues compared to those the navigation policy was trained on. This makes it difficult to isolate the VLM's memory contribution in the end-to-end system.
>
>
> Thank you for raising this concern. We believe there is a misunderstanding of our evaluation protocol. Our eval design intentionally separates the VLM’s memory performance from the navigation controller through two distinct metrics. The **high-level metrics** (HL-SR / HL-SPL) evaluate the VLM only at the level of frame selection: as described in Appendix Sec. B.3, we teleport the agent to the pose of the predicted frame and check geometric and semantic satisfaction criteria, and HL-SPL is computed using cached pose distances. These metrics are completely independent of the low-level policy and directly measure the VLM’s memory recall and reasoning quality.
>
>
> The **low-level metrics** (LL-SR / LL-SPL) are defined (conditionally) on successful high-level predictions and capture the navigation controller’s ability to reach the selected target efficiently. This separation is precisely meant to avoid conflating VLM memory quality with distribution shift or navigation failures. End-to-end LL performance is still meaningful, since embodied memory involves both selecting the right goal and navigating to it, but the benchmark cleanly disentangles these two factors.
>
> ---------
> ### Confidence Intervals
> > The paper reports all key results as point estimates (e.g., Figure 2) without confidence intervals or standard deviations from multiple runs. While the high cost of full evaluations is acknowledged, this lack of uncertainty reporting also extends to smaller-scale analyses like the video length subsampling study and the cross-benchmark transfer experiments, where multi-run statistics might have been more feasible.
>
> We agree that reporting uncertainty is valuable, and we attempted multi-seed evaluations for the OSS models during development (using their recommended temperature values for sampling). However, across repeated runs we observed **negligible variation (<1%)**, largely because the evaluation pipeline and baselines are deterministic once the instruction and episode logs are fixed. This consistency held both for the main results and for the analyses referenced by the reviewer.
>
> ---------
> ### Alternate Embodiments
> > Do authors plan to include diverse embodiments beyond the Fetch robot?
>
> We would like to clarify that our benchmark currently uses the Stretch robot embodiment. The underlying simulator supports multiple embodiments (e.g., Fetch, Spot), so extending the benchmark to additional robot platforms is fully feasible, but we view this as a direction for future work.

---

### Official Review · Reviewer_Akdb · 2025-10-27

**Soundness:** 2
**Presentation:** 3
**Contribution:** 3
**Rating:** 4
**Confidence:** 4

**Summary:**

This paper introduces a new benchmark designed to evaluate long-horizon memory in embodied agents.

The benchmark operates in the photorealistic Habitat simulator and is structured in a two-phase process: an initial experience collection phase where an oracle agent performs a series of pick-and-place interactions over a long trajectory, followed by an interaction phase where a test agent is given the complete history (video, poses, actions) and must complete memory-dependent tasks. The paper argues that this setup effectively isolates memory capabilities from exploration challenges.

FINDINGDORY features 60 diverse, procedurally generated task templates that probe spatial, temporal, and conditional reasoning. The authors evaluate a range of state-of-the-art Vision-Language Models (VLMs) within a hierarchical policy framework, where a high-level VLM selects a goal frame from the history and a low-level policy navigates to it. The results demonstrate that current VLMs, even powerful closed-source models, struggle significantly with these tasks, particularly those involving multi-hop temporal reasoning or multiple sequential goals.

**Strengths:**

(1) The paper addresses a clear and critical gap in current research: the lack of rigorous benchmarks for long-horizon memory in embodied agents. The authors convincingly argue why existing video QA and embodied QA benchmarks are insufficient. The problem it tries to solve is of great significant.

(2) I think the two-phase setup that decouples experience collection from the evaluation phase is a very strong methodological choice. This design effectively isolates an agent's memory and reasoning capabilities from its exploration skills, allowing for a more controlled and focused evaluation.

(3) The paper provides a comprehensive analysis of modern VLMs, revealing their significant shortcomings in long-context embodied reasoning. The finding that performance often degrades with more frames for frozen VLMs, and that all models struggle with multi-goal and fine-grained spatial tasks, offers valuable insights to the community.

**Weaknesses:**

(1) My main reservation is that the evaluation is tightly coupled with a specific hierarchical policy (VLM for goal-frame selection + navigation controller). The paper itself shows in Figure that the performance of this system drops massively when the low-level navigation policy is introduced. This makes it difficult to disentangle the source of failure. For example, is a low success rate on a task due to the VLM's inability to recall the correct information, or is it because the VLM correctly identified a goal, but the chosen frame provided poor visual cues for the navigation policy, causing it to fail? I think this confounding factor complicates the claim that the benchmark purely evaluates memory.

(2) Moreover, I wonder if the current evaluation truly measures memory in an agent-centric sense (i.e., the ability to build and maintain a compressed, internal world state) or if it primarily tests the long-context retrieval capabilities of VLMs on a single, massive prompt. The paper criticizes needle-in-a-haystack tasks but the baseline setup, which feeds the entire video history to the VLM at once, feels conceptually similar.

**Questions:**

(1) Could you please elaborate on the significant performance drop between HL-SR and LL-SR? How can we be confident that the benchmark results reflect the memory limitations of VLMs, rather than the fragility of the chosen low-level controller or the distribution shift between its training data and the VLM-selected goal frames?

(2) How does the FINDINGDORY setup, particularly for the evaluated baselines, differentiate itself from long-video QA tasks that test retrieval from a long context? Have you considered alternative agent architectures that would force the model to build an explicit and evolving memory representation, rather than processing the full history at each step? （If the answer is already in the appendix and I overlook it (this might happen) , please also kindly point it out）

---

> ### Author Response · Authors · 2025-11-24
> **Rebuttal**
>
> ### Clarification on Decoupled Evaluation Strategy
> >My main reservation is that the evaluation is tightly coupled with a specific hierarchical policy (VLM for goal-frame selection + navigation controller). The paper itself shows in Figure that the performance of this system drops massively when the low-level navigation policy is introduced. This makes it difficult to disentangle the source of failure. For example, is a low success rate on a task due to the VLM's inability to recall the correct information, or is it because the VLM correctly identified a goal, but the chosen frame provided poor visual cues for the navigation policy, causing it to fail? I think this confounding factor complicates the claim that the benchmark purely evaluates memory.
>
> >Could you please elaborate on the significant performance drop between HL-SR and LL-SR? How can we be confident that the benchmark results reflect the memory limitations of VLMs, rather than the fragility of the chosen low-level controller or the distribution shift between its training data and the VLM-selected goal frames?
>
>
> Thank you for raising this concern. We believe there is a misunderstanding about our evaluation protocol.
>
>
> Our high-level policy evaluation is defined at the level of frame selection, not at the level of the navigation controller. As described in Appendix Sec. C.3 (lines 865-866), the High-Level Success Rate (HL-SR) is computed by teleporting the agent in the simulator to the pose associated with the frame(s) predicted by the VLM and then checking satisfaction of geometric and semantic criteria (defined in lines 854-863). Similarly, HL-SPL computes the SPL metric (defined in lines 875-879) where distance to the closest solution frame from the agent current position is computed from the cached pose information. Thus, HL-SR/HL-SPL directly measures how effective the VLM was at selecting the correct solution frame. More importantly, this metric is completely independent of the behavior of low-level navigation policy.
>
>
> The Low-Level Success Rate (LL-SR) and LL-SPL are intentionally defined conditionally on HL-SR (lines 869-871). They quantify, in isolation, whether a navigation controller can reach the target entity and does it do so efficiently, given a high-level frame that is already deemed successful for task completion. This separation is precisely meant to disentangle VLM’s “memory recall and reasoning capability” (captured by HL-SR/HL-SPL) from the embodied navigation challenges that arise once a good goal is provided - since the low level task could also benefit from memory when finding the shortest path to a goal (captured by LL-SR/LL-SPL).
>
>
> **We attribute the large drop in SR to distribution shift rather than poor navigation policy quality.** In isolation, our ImageNav policy is not a weak controller. As reported in line 1413, the policy achieves a **78% success rate** and **48.21 SPL** on 1000 validation episodes using standard image-goal navigation tasks derived from our validation scenes. This shows that the policy is well-behaved when the goal images resemble the distribution it was trained on. The performance drop in LL-SR occurs specifically when the controller is asked to navigate to **VLM-selected goal frames**, many of which come from phases of the episodes (e.g., pick/place interactions, unusual agent viewpoints) that are out of distribution for the ImageNav policy. We argue this drop is not a confounder in the evaluation itself, but an informative indication that high-level policy to low-level policy hand-off is a bottleneck in hierarchical embodied memory mechanisms.
> We also highlight that this disentanglement allows researchers to decide whether they would like to study memory of an isolated component, or integrated with a controller (which is also important for practical deployment). This flexibility is a strength of the benchmark. We will clarify these evaluation details in the revision, emphasize the teleportation-based HL evaluation, and explicitly report the isolated ImageNav performance to avoid misunderstanding.
>
>
> ---------
> ### Long-context Retrieval Clarification
> >Moreover, I wonder if the current evaluation truly measures memory in an agent-centric sense (i.e., the ability to build and maintain a compressed, internal world state) or if it primarily tests the long-context retrieval capabilities of VLMs on a single, massive prompt.
>
> Thank you for raising this point. Our intention is not to enforce a specific notion of “memory’’ but to create tasks where success inherently requires recalling information from earlier interactions, regardless of whether a model uses compressed internal state, explicit external state, summarization, or long-context retrieval. While our baseline feeds the full history to a VLM, this reflects current architectural constraints of the state of the art foundation models, not a methodological requirement.

---

> ### Author Response · Authors · 2025-11-24
> **Rebuttal**
>
> ### Needle-in-a-Haystack Eval Critique
> > The paper criticizes needle-in-a-haystack tasks but the baseline setup, which feeds the entire video history to the VLM at once, feels conceptually similar.
>
> Our critique of needle-in-a-haystack (NiH) tasks is not that they require locating a relevant frame in a long video. The issue is that NiH tasks typically construct the “needle’’ in a highly synthetic way such that the problem reduces to a single key-value lookup, rather than complex long-horizon reasoning.
>
> In contrast, the high-level task in FindingDory requires looking at a sequence of information to decide goals for navigation. The relevant information is embedded within realistic, interaction-driven trajectories, and the agent often must piece together multiple events like object rearrangements, temporal dependencies, distractor interactions, and the current sub-goal it is attempting to solve.
>
> ---------
> ### Baselines and Alternate Memory Architectures
> >How does the FINDINGDORY setup, particularly for the evaluated baselines, differentiate itself from long-video QA tasks that test retrieval from a long context? Have you considered alternative agent architectures that would force the model to build an explicit and evolving memory representation, rather than processing the full history at each step? （If the answer is already in the appendix and I overlook it (this might happen) , please also kindly point it out
>
> **Clarification about the baselines:**
> Our VLM-based baselines do not process the entire history at every step. The history is encoded once during goal-frame extraction; navigation policies then operate only on the selected frames.
>
>
> **Alternative architectures:**
>
> We tried exploring alternate approaches that build explicit, evolving memory. Our Textual Memory Agent (lines 287-291) builds a temporally structured memory by chunking the video and summarizing each segment before an LLM reasons over the resulting textual history. Despite this, it performs the worst, suggesting that lightweight summaries lose essential spatial-temporal detail required for embodied tasks. Based on reviewer NYTL suggestions, we are now implementing a stronger baseline using 3D-Mem [1], which constructs a structured spatio-temporal memory in 3D space. We will post the results as soon as we are able to get it working.
>
> Importantly, this is precisely the type of research FindingDory is designed to enable: developing and evaluating alternative memory architectures for long-horizon embodied tasks, whether through long-context VLMs, summarization-based agents, or explicit 3D memory systems.
>
> [1] 3D-Mem: 3D Scene Memory for Embodied Exploration and Reasoning.

---

### Official Review · Reviewer_NYTL · 2025-10-29

**Soundness:** 3
**Presentation:** 3
**Contribution:** 2
**Rating:** 4
**Confidence:** 3

**Summary:**

This paper presents FINDINGDORY, a benchmark developed to evaluate the long-horizon memory capabilities of agents within the Habitat simulator. Its primary focus is on assessing long-range spatiotemporal reasoning that relies solely on memory, while mitigating confounding factors particularly associated with exploration. The benchmark comprises 60 diverse navigation tasks, categorized along dimensions of memory requirements. It integrates dynamic environments, which necessitate agents to reason over evolving contexts, and supports procedural extensibility—allowing task complexity to scale in tandem with advancements in embodied agents. The paper’s main contributions are threefold: 1) the creation of 60 diverse navigation tasks that demand spatiotemporal reasoning; 2) the evaluation of high-level policies combined with low-level navigation policies in memory-intensive scenarios; and 3) the derivation of insights to inform the development of memory-efficient embodied agents.

**Strengths:**

1. The design of navigation tasks emphasizes that agents rely exclusively on past interaction information for high-level goal selection. This explicitly assesses VLMs' memory retrieval capabilities across diverse dimensions, including single-target spatial tasks, single-target temporal tasks, and multi-target tasks.
2. The integration of high-level and low-level policies reveals that different low-level navigation skills lead to varying degrees of task completion when performing memory-based practical navigation.
3. Tasks can be progressively extended in complexity in accordance with the evolving capabilities of VLMs.

**Weaknesses:**

Oracle agents were employed during memory collection, which appears to introduce several stringent assumptions:
1. The memory construction assumes that all consecutive subtasks are successfully completed, and each is executed in the most efficient manner, i.e., via the shortest path. The experiences collected through this method are excessively "clean" and inconsistent with real-world scenarios. This is because completing multiple subtasks is highly challenging—evidenced by a mere 26.4% success rate on the GOAT-Bench [1]. This assumption restricts the approach’s applicability in complex, real-world memory settings.
2. All memories are stored in video format. Due to the upper limit on the number of images that VLMs can process, downsampling becomes necessary during memory retrieval. However, frame-rate-based downsampling has been shown in Mobility VLA [2] experiments to cause information loss, leading to a certain degree of performance degradation—particularly in the processing of small objects. This seems to introduce a level of unfairness in the experimental setup (lines 82-83). A more efficient memory storage method should perhaps be adopted to standardize the input image sets across different VLMs, such as using informative snapshot images to store key information, as implemented in 3D-Mem [3].

[1] GOAT-Bench: A Benchmark for Multi-Modal Lifelong Navigation.
[2] Mobility VLA: Multimodal Instruction Navigation with Long-Context VLMs and Topological Graphs.
[3] 3D-Mem: 3D Scene Memory for Embodied Exploration and Reasoning.

Typo: Line 818: "the and effector" -> "the end effector"

**Questions:**

1. What exactly is the distance threshold for success determination? In lines 426-427 of the paper, a threshold of 1.0 meter is selected for receptacles; however, in lines 855-856, the thresholds are specified as 2.0 meters for objects and 0.1 meters for receptacles.
2. Although both SPL and SR are expressed as percentages, they describe distinct aspects of navigation performance. Combining them in a single visualization, as done in Figures 3c and 3d, makes the comparison rather unintuitive.
3. It is suggested to add an analysis of failure causes for high-level retrieval, particularly by comparing the performance of Qwen before and after fine-tuning. This analysis would clarify the specific capabilities enhanced through the fine-tuning process.
4. In Figure 4a, the fine-tuned Qwen demonstrates a trend where success rate increases with video length. However, the experimental results are truncated at 96 frames. Could extending the video length further improve the success rate? Additionally, the input limits of different VLMs should be explicitly indicated in this figure.
5. The paper repeatedly mentions the advantage of visual realism. It is recommended to supplement with real-robot experiments to demonstrate that the fine-tuned Qwen exhibits a smaller sim-to-real gap in terms of visual perception.

---

> ### Author Response · Authors · 2025-11-24
> **Rebuttal**
>
> ### Excessively Clean Experience Trajectories
>
> > The experiences collected through this method are excessively "clean" and inconsistent with real-world scenarios. This is because completing multiple subtasks is highly challenging—evidenced by a mere 26.4% success rate on the GOAT-Bench [1]. This assumption restricts the approach’s applicability in complex, real-world memory settings
>
>
> We appreciate the reviewer’s point and agree that real-world robot trajectories often contain failed or partial attempts. However, we note that:
>
> 1. Adding realistic noise would only increase the difficulty of the memory problem. Current models already struggle even under clean conditions, so our setup represents a conservative lower bound rather than an unrealistic simplification.
> 2. Most of our tasks require looking at a subset of interactions to reach a solution, so the other interactions act as noise for the particular task.
> 3. Our tasks are not limited to “retrying a previously successful action’’; many require recalling incidental observations made while performing other interactions. Thus these interactions could be seen as noise w.r.t the current instruction (e.g., Navigate to an apple you did not interact with).
>
> Overall, our generation framework is very flexible, and we can incorporate such things in an updated benchmark once researchers make progress on the current version of FindingDory.
>
> ------
>
> ### Naive Frame-Rate based Downsampling
> > However, frame-rate-based downsampling has been shown in Mobility VLA [2] experiments to cause information loss, leading to a certain degree of performance degradation—particularly in the processing of small objects. This seems to introduce a level of unfairness in the experimental setup (lines 82-83). A more efficient memory storage method should perhaps be adopted to standardize the input image sets across different VLMs, such as using informative snapshot images to store key information, as implemented in 3D-Mem [3].
>
> We thank the reviewer for highlighting this issue and for pointing us to 3D-Mem. We agree that frame-rate based downsampling can cause information loss. To mitigate this, we evaluated multiple VLMs over a range of input frame budgets and reported its best performance. Empirically, we found that performance is often highest at intermediate context lengths rather than at the maximum the model can accept. This suggests that current long context VLMs are not yet able to reliably exploit more visual history simply by increasing the number of frames, and that benchmarks like FindingDory can help reveal these limitations when comparing models.
>
> We also agree that more efficient and principled storage schemes are also an important research direction and 3D-Mem is exactly one of the kinds of research we hope FindingDory will stimulate by providing a standardized, challenging testbed for memory dependent tasks. We are currently implementing 3D-Mem in our codebase and we will include the corresponding results in the rebuttal as soon as we have the results.

---

> ### Author Response · Authors · 2025-11-24
> **Rebuttal**
>
> ### Clarification on Success Thresholds
> > What exactly is the distance threshold for success determination? In lines 426-427 of the paper, a threshold of 1.0 meter is selected for receptacles; however, in lines 855-856, the thresholds are specified as 2.0 meters for objects and 0.1 meters for receptacles.
>
>
> Thank you for pointing this out. To clarify: the distance threshold for success determination for receptacle is 0.1 m in all FindingDory evaluation episodes (see App B.3; line 901). The two distance thresholds linked in lines 429 of the updated manuscript are purely for diagnostic analysis of the low-level image navigation policy and do not relate to the FindingDory task success rates. We provide detailed clarification in the following.
>
>
> **Image Navigation Policy Success Criteria.** As noted in App F.1, we use an independent success criteria when training the low-level image navigation policy. This criteria is defined in line 1351 of the manuscript which states that the navigation policy is trained to reach within 1.0 m of the goal image position which is standard formulation in prior work [2]. We note that this success criteria is not used in FindingDory evaluations in any way for success determination and only used during RL training of the navigation policy.
>
>
> **Clarification on lines 426-427.** In these lines, our objective was to diagnose the failure modes of the low level navigation policy. Hence we created a relaxed threshold of 2.0 m to analyze if the low level navigation policy atleast reaches the vicinity of the VLM selected goal frame or just fails catastrophically at much larger distances. We incorrectly stated “1.0 m, in case of receptacles”. Instead, the intention was to create a threshold of 2.0 m that is more lenient in comparison to the success threshold that the navigation policy is trained for, which is 1.0 m.
>
>
> We have updated the paper to make the success determination process more clear (lines 429-431 and 934-950).
>
>
> [1] HomeRobot: Open-Vocabulary Mobile Manipulation, CoRL 2023.
>
> [2] OVRL-V2: A simple state-of-art baseline for ImageNav and ObjectNav, arxiv 2023.
>
>
> ---------
> ### SR/SPL Plot Cleanup
> > Although both SPL and SR are expressed as percentages, they describe distinct aspects of navigation performance. Combining them in a single visualization, as done in Figures 3c and 3d, makes the comparison rather unintuitive.
>
>
> We grouped SR and SPL in the same plot primarily due to (1) space constraints and (2) the fact that the two metrics are tightly related. Although they measure different aspects of performance, SPL is always upper-bounded by SR and captures the efficiency of successful trajectories. Keeping them together makes this relationship immediately visible. That said, we have added separate SR and SPL plots in the Appendix (Fig 18) and we are happy to move it to the main paper in the camera-ready if the reviewer suggests.
>
>
> ---------
> ### Finetuned Qwen vs Zero-Shot Qwen Failure Analysis
>
> > It is suggested to add an analysis of failure causes for high-level retrieval, particularly by comparing the performance of Qwen before and after fine-tuning. This analysis would clarify the specific capabilities enhanced through the fine-tuning process.
>
> We analyse the failure modes for most high-level baselines in Figure 18-23 in the appendix. As per reviewer’s suggestion, we have added detailed task-wise High-Level Success Rate (HL-SR) plots in Appendix G of the manuscript (see Fig. 16). Across all 11 task categories, the supervised fine-tuned Qwen2.5-VL-3B model consistently and substantially outperforms the zero-shot Qwen2.5-VL-7B baseline, suggesting that VLM finetuning is crucial for improved performance on the long-horizon, memory-focused FindingDory task suite. While both models experience performance degradation as the number of pick-and-place interactions increase, the SFT model exhibits stable performance trends, preserving meaningful accuracy across tasks that require duration tracking, object/timestamp-based recall and interaction ordering. In contrast, the zero-shot model performance collapses rapidly, often approaching zero beyond 5–8 interactions on multi-goal and spatial relationship tasks. Although the SFT model does well on recall-based tasks, its performance on multi-goal (revisitation-based), conditional interaction and spatial relationship tasks drops significantly as the interaction video length increases. This suggests that even though full fine tuning of the VLM leads to improved performance, it does not endow the model with truly long-horizon multi-hop reasoning which highlights the need for developing better memory and reasoning architectures.

---

> ### Author Response · Authors · 2025-11-24
> **Rebuttal**
>
> ### Finetuning Qwen beyond 96 Frames
> > In Figure 4a, the fine-tuned Qwen demonstrates a trend where success rate increases with video length. However, the experimental results are truncated at 96 frames. Could extending the video length further improve the success rate? Additionally, the input limits of different VLMs should be explicitly indicated in this figure.
>
> Thank you for the suggestion. We agree that understanding how performance scales with longer video histories is an important research question. Prior work on long-form video understanding (for example LLava-Video[1], MovieChat [2]) has shown that accuracy indeed improves as more frames are provided, typically only up to a point where gains begin to plateau or drop as more frames are provided. Our benchmark is designed to facilitate exactly this kind of analysis for embodied memory, and we see this as a valuable direction for future work.
>
> In our current experiments, we were only able to scale histories up to 96 frames due to compute constraints in an academic setting. Our fine-tuning setup uses 8 A40 GPUs with 48 GB each with a batch size of 1 per single GPU, training for 96-frame inputs requires approximately 5 days. Larger sequence lengths grow memory quadratically and become prohibitively expensive. In response to the reviewer’s comment, we attempted to extend sequence length using context parallelism (CP). However, based on our initial investigation, the current implementations in HuggingFace do not yet support CP with vision language models. Even if we were to get it to work, the use of 2 GPUs for 1 batch would mean that our training time would double to around 10 days, making it infeasible for us at present.
>
> [1] LLaVA-Video: Video instruction tuning with synthetic data, TMLR 2025.
>
> [2] MovieChat: From Dense Token to Sparse Memory for Long Video Understanding, CVPR 2024
>
> ---------
> ### Sim-to-Real Gap Clarification
> > The paper repeatedly mentions the advantage of visual realism. It is recommended to supplement with real-robot experiments to demonstrate that the fine-tuned Qwen exhibits a smaller sim-to-real gap in terms of visual perception.
>
> We note that although we do not currently present dedicated real-robot experiments, our cross-benchmark evaluation (Lines 458–473) provides a proxy measure of sim-to-real generalisation: co-training on FindingDory improves performance on the real-world video-scan dataset VSI‑Bench (Table 3), demonstrating reduced gap when transferring from our simulator to a real-expected environment.
> There is also some prior work that shows that visual policies trained in the Habitat simulator can transfer to the real world for the ImageNav task when coupled with pretrained visual representations [1]. While full real-robot verification remains future work, the literature suggests that the visual fidelity and procedural variability of our simulation setup contribute to more robust transfer.
>
> [1] What Do We Learn from a Large-Scale Study of Pre-Trained Visual Representations in Sim and Real Environments?, ICRA 2024

---

### Official Review · Reviewer_AukX · 2025-11-01

**Soundness:** 3
**Presentation:** 3
**Contribution:** 3
**Rating:** 6
**Confidence:** 4

**Summary:**

The manuscript proposes a benchmark focused on challenging and evaluating embodied AI agents memory-intensive tasks within a virtual environment. Among the manuscripts claims are that the proposed benchmark is best-suited for evaluating long-horizon tasks, provides a comprehensive evaluation for VLM-based high-level (goal-prediction) policies as well as low-level navigation policies, and provides new effective and extensible metrics for studying/developing memory-efficient agents.

**Strengths:**

- The manuscript is well-written and well-organized
- The manuscript considers compelling tasks and reasoning mechanisms in Embodied AI
- The manuscript provides a reasonable set of initial experiments that provide sufficient headroom for subsequent research
- The paper provides a good amount of experiments

**Weaknesses:**

The manuscript is missing a principled discussion of why the tasks were generated in the way that they were. Why were the memory tasks generated according to the templates in Section 3.1, specifically? How was it ensured in the task design that these tasks are meaningful in some way, e.g., resemble naturally-occurring tasks?

**Questions:**

Nothing additional; please see questions above.

---

> ### Author Response · Authors · 2025-11-24
> **Rebuttal**
>
> ### Relation between Task Templates and Memory Tasks
> >The manuscript is missing a principled discussion of why the tasks were generated in the way that they were. Why were the memory tasks generated according to the templates in Section 3.1, specifically?
>
> First, we want to clarify why we use templates:
> * Templates let us procedurally sample many diverse environment configurations while still supporting automatic correctness checks via a symbolic validator (based on PDDL), which is essential at the scale of our benchmark.
> * Templates also give us procedural extensibility: for many task families, we can increase difficulty simply by lengthening the experience collection phase or increasing the number of object interactions, and the same template then induces a longer horizon and more complex temporal reasoning problem.
>
> The particular templates in Table 5 were selected after extensive iteration to cover a broad range of memory demands, including:
> * Spatial Recall: How were objects and rooms located spatially with respect to each other and the robot.
> * Interaction Awareness: Which objects and receptacles were interacted with and which spaces were visited.
> * Semantic Generalization of Memory: Can we refer to past events indirectly using new attributes?
> * Temporal Recall: Which events happened in what order and for how long
>
> Through these holistic sets of subtasks, we also explicitly avoided assumptions that enable narrow hand-crafted solutions, such as static scenes with fixed instruction vocabularies (which semantic mapping approaches exploit) or the assumption that a purely text based memory is expressive and complete in a control setting.
>
> Task construction and verification were among the most technically involved aspects of the benchmark, and we will revise the paper to make this design rationale and validation process more explicit. Thank you for raising this. If the reviewer is satisfied with our response, we will add it to the manuscript.
>
>
> ### Relevance of Task Design
> >How was it ensured in the task design that these tasks are meaningful in some way, e.g., resemble naturally-occurring tasks?
>
> Many everyday requests that humans would want their robots to do will implicitly reference past interactions, such as:
> * *“Can you find the notebook you were moving when you cleaned up yesterday?”* (Table 2: Conditional Interaction)
> * *“I asked you to put my jacket away in the evening, can you bring it back?”* (Table 2: Time Based)
> * *“There were several books on this table yesterday; where did you keep the third one?”* (Table 2: Interaction Order)
> * *“Check whether you left the keys on any of the tables you interacted with.”*  (Table 2: Unordered Revisitation)
>
> These scenarios map directly onto the core operations encoded in our task templates, such as remembering where objects were placed, recalling the sequence of earlier manipulations, and revisiting locations tied to past actions.
>
> Our design philosophy was to abstract familiar robot instructions into structured forms that allow controlled memory evaluation while preserving the essential episodic and temporal reasoning demands present in natural tasks. Even though the benchmark uses templated instructions to ensure reproducibility and systematic scaling, we deliberately incorporated real-world complexities such as object rearrangements, visually similar distractors, and multi-step causal dependencies. These elements ensure that solving the tasks requires nontrivial memory use rather than pattern matching or shortcut exploitation.

---

> > ### Comment · Reviewer_AukX · 2025-11-27
> >
> > The authors adequately satisfy my main concerns about task design. Yes, I encourage the authors to add this discussion to the manuscript.
> >
> > I will keep my current score.

---

### Author Response · Authors · 2025-11-24
**Rebuttal Response**

We would like to thank all the reviewers for their detailed and helpful feedback. The reviewers consistently agree that the paper tackles an important and underexplored problem: evaluating long-horizon memory in embodied agents. **Akdb** calls this a “clear and critical gap” in current research, and **63kF** similarly notes that the work addresses a “crucial and underexplored area,” with a compelling motivation distinguishing it from static video QA benchmarks.

The benchmark design receives strong praise across reviewers. **NYTL, Akdb,** and **63kF** highlight the strength of the 60 diverse, procedurally generated task templates that probe spatial, temporal, and multi-goal reasoning. **NYTL** emphasizes that tasks force agents to “rely exclusively on past interaction information,” while **Akdb** calls the two-phase setup a “very strong methodological choice” that isolates memory from exploration. **63kF** further notes the benchmark’s extensibility and well-defined success criteria.
The evaluation methodology is also viewed positively. **AukX** describes the manuscript as “well-written and well-organized,” with a “good amount of experiments.” **63kF** commends the “well-specified, multi-faceted metrics” (HL-SR/SPL, LL-SR/SPL, DTG-SR, SC-SR), and **Akdb** highlights the comprehensive analysis of modern VLMs.

Finally, the empirical findings are seen as valuable to the community. **Akdb** and **63kF** emphasize that even strong models (e.g., GPT-4o, Gemini-2.0) struggle, especially on multi-goal and temporal tasks, providing actionable insights for developing more memory-efficient embodied agents.


## Summary of Changes

We address all concerns below and have updated the manuscript accordingly. Key revisions include:

• **Lines 308–315:** Clarified what we meant by “unsolvable’’ tasks in FindingDory.

• **Figure 2:** Added the frame budget used per VLM in high-level evaluations.

• **Lines 417–421:**  A small clarification on why we believe that the reason for the drop in performance for the hierarchical ImageNav agent is the distribution shift of the goal images.

• **Lines 429–431:** Clarified the distance threshold used for the low-level navigation success criteria.

• **Lines 934–950:** Expanded the success criteria description in the Appendix.

• **Lines 1458–1511:** Added a detailed failure analysis comparing Qwen-SFT and Qwen-Zero-Shot.

• **Lines 1512–1522:** Added extended hierarchical policy performance plots with separate SR and SPL views.

---

> ### Author Response · Authors · 2025-12-03
> **Global Rebuttal Summary**
>
> Dear ACs, SACs, PCs,
>
> Since there will be no author-reviewer discussion phase, we provide a summary of the high-level concerns we addressed, to assist your final decision.
>
> Task template design (**AukX**)
>
> * **Concern**: Explanation of why tasks use the specific templates in Section 3.1 and how they relate to naturally occurring robot instructions.
> * **Resolution**: We clarify that templates are needed for (i) procedural diversity and (ii) automated correctness checking via a PDDL-based validator at scale. The final set was chosen after extensive iteration to cover different kinds of memory-based abilities, which prevent shortcut solutions like closed-set vocabularies or relying on text-only memories. The reviewer noted their concerns were adequately addressed and **maintained a positive score of 6**.
>
> For the following concerns, we did not get a chance to engage with the reviewers.
>
> Trajectory realism (**NYTL**)
>
> * **Concern**: Oracle-generated trajectories are unrealistically “clean”.
> * **Resolution**: Clean trajectories make the benchmark a lower bound on difficulty, since adding failures and retries would only make memory harder and current models already struggle. Also, our videos typically have multiple interactions, some of which function as structured noise for a given instruction. For “magic grasp,” we note that we orient the robot, visually center the object, and move the arm close before grasping, creating a short, interpretable interaction segment; videos on our website illustrate this (https://findingdorybenchmark.github.io/).
>
> Magic grasp is unrealistic (**63kF**)
>
> * **Concern**: “magic grasp” actions introduce non-natural temporal transitions which will be hard for models to recognize.
> * **Resolution**: To make the interaction more realistic we orient the robot, visually center the object, and move the arm close before grasping, creating a short, interpretable interaction segment; videos on our website illustrate this (https://findingdorybenchmark.github.io/).
>
> Video downsampling (**NYTL**)
>
> * **Concern:** Storing memory as videos and using frame-rate downsampling introduces unfair information loss
> * **Resolution:** We agree that naive downsampling can lose detail and this is why we evaluated each VLM over multiple frame budgets, reporting the best performance per model in Fig 2. We also **added experiments using the 3D-Mem approach** as suggested below this comment.
>
> Qwen2.5-VL failure analysis (**NYTL**)
>
> * **Concern:** More explicit analysis of how fine tuning changes Qwen’s behavior.
> * **Resolution:** We added detailed category wise success plots comparing Qwen2.5-VL-3B (fine tuned) and Qwen2.5-VL-7B (zero shot) in the App. Sec G of paper. Fine tuning consistently improves performance across all tasks, although both models still degrade on long multi-goal and spatial tasks, underscoring the difficulty of our benchmark.
>
> Memory vs retrieval (**Akdb**)
>
> * **Concern:** Does benchmark mainly test long-context retrieval on a single prompt?
> * **Resolution:** We argue that, unlike synthetic needle in a haystack task, FindingDory embeds evidence in realistic interaction trajectories with object rearrangements, distractors, and temporal dependencies, often requiring integration of multiple events to reach a final answer.
>
> Unsolvable tasks (**63kF**)
>
> * **Concern:** “unsolvable” instructions are always counted as failures without an abstain option.
> * **Resolution:** We clarify that “unsolvable” refers to the limitations of the current frame-selection-based goal parameterization of the baselines representing state-of-the-art currently: some instructions become impossible when no frame satisfies the goal criteria.
>
> Attribute annotations (**63kF**)
>
> * **Concern:** Attribute tasks rely on GPT-4o descriptions only in validation
> * **Resolution:** Attribute tasks appear only in validation because train and val use disjoint object sets and annotating the entire training set would be costly; all GPT-4o attributes in validation were manually checked to ensure correctness and to reduce bias. These tasks also act as held-out evals that test generalization beyond the training templates.
>
> Sim to real transfer (**NYTL**)
>
> * **Concern:** Evidence supporting the claim about benchmark’s realism reducing sim to real gap
> * **Resolution:** We report cross benchmark results in Table 3, where co-training on FindingDory improves performance on VSI Bench, a real video scan dataset, suggesting transfer to real imagery.

---

> > ### Author Response · Authors · 2025-12-03
> > **Global Rebuttal Summary**
> >
> > Navigation bottleneck in FindingDory (**NYTL**, **Akdb**, **63kF**)
> >
> > * **Concern:** Large drop from High Level Success Rate (HL-SR) to Low Level Success Rate (LL-SR) confounds VLM memory with navigation failures.
> > * **Resolution:** High level metrics are computed by teleporting the agent to predicted frame poses and using cached distances, completely independent of any controller. Low level metrics are defined conditionally on successful high level predictions and only test the controller’s ability to reach the target efficiently (which also improves with memory). The ImageNav policy achieves a good success rate when evaluated standalone on standard ImageNav tasks, so the LL drop is attributed to distribution shift between its training distribution and VLM-selected goal frames.
> >
> > ## **Requested Experiment: 3D-Mem Baseline Results**
> >
> > As per reviewer Akdb and NYTL suggestion, we implemented the 3D-Mem \[1\] baseline that incrementally builds an efficient, structured 3D representation of the environment. We provide a description of the method and results in the following.
> >
> > **Method Overview.** 3D-Mem compresses the input video sequence by selecting a single representative frame (termed a *memory snapshot*) for each cluster of co-visible object instances, rather than storing all frames in the sequence. This is accomplished through hierarchical clustering over detected object instances across frames. A frame is assigned to a cluster when it provides full visibility of all objects within that cluster; if no such frame exists, the cluster is recursively subdivided via K-Means applied to the 2D object locations. This procedure yields a compact set of memory snapshots that collectively preserve visibility of all salient objects. During inference, a “prefiltering” stage retrieves relevant memory snapshots by asking an LLM to select the most relevant object categories from all unique categories detected in the full video. Then, all memory snapshot images corresponding to “K” object categories from the relevant category set (we use K=10 as original paper) are used as in-context images for the final VLM call.
> >
> > **Results and Analysis.** We evaluate the 3D-Mem baseline on the FindingDory task suite by supplying images selected by 3D-Mem in the input context window of the VLM. This is in contrast to uniformly subsampling the entire video sequence collected by the oracle agent. In our evaluations, we find that 3D-Mem only achieves a HL-SR of 5% on the full task suite which is a significant drop from the 15.14% HL-SR obtained by uniform subsampling to 96 frames. We attribute this drop to the following:
> >
> > 1. 3D-Mem manages memory only in an object-centric manner which does not account for the dynamic object rearrangement throughout the video sequence when pick-place actions are executed by the oracle agent. As a result, many of the “interaction-based” FindingDory tasks cannot be solved as the relevant frames showcasing dynamic pick–place interactions are dropped during the co-visibility clustering routine.
> > 2. The “prefiltering” routine to retrieve relevant memory snapshot images from the constructed memory relies on semantic co-occurrence to select the reduced set of relevant object categories. But FindingDory tasks are designed to counter such prior knowledge-based solutions as objects (and distractors) are spawned randomly and rearranged during video collection (lines 798-804 of paper) .
> > 3. 3D-Mem performs co-visibility clustering based on the 2D positions of objects to generate new candidate cluster centers. This simple heuristic method does not ensure that the frames selected as memory snapshots are guaranteed to have the target objects at a “close-enough” distance to satisfy FindingDory task success thresholds (lines 896-909 of paper). We observe that in many cases 3D-Mem is able to select the correct frame but the target objects are far away from the agent perspective.
> >
> > Overall our evaluations suggest that future methods should try to build efficient memory management solutions that account for dynamic scene rearrangements with better memory retrieval mechanisms.
> >
> > \[1\] 3D-Mem: 3D Scene Memory for Embodied Exploration and Reasoning.

---

### Meta-Review · Area_Chair_HPm2 · 2026-01-11

**Summary:**

The paper's merit is on evaluating long-horizon memory in embodied agents, and all reviewers agree that it is a reasonable design for two-phase setup isolating memory from exploration, providing VLM evaluation with actionable insights. There are still unresolved concerns mainly on the realistic improvement using this data, like no real-world robot validation, using clean oracle trajectories (though the argument is that it is a lowerbound), etc. Given these concerns, I am recommending a rejection but would not mind if the paper gets accepted.

**Reviewer Concerns:**

A useful result in the rebuttal is the added new results on 3D-Mem, which will be good to be incorporated into the final version. One main argument from authors about the dataset using overly clean oracle trajectories is that  they represent a lower bound, which makes sense but not the ideal way. Thus,  sim-to-real transfer evidence given unrealistic success assumption is the major concern. Authors reported cross-benchmark results showing co-training on FindingDory improves performance on VSI-Bench (real video dataset), demonstrating transfer. However, VSI-Bench is not for real-world robotics, testing visual-spatial reasoning like absolute distance estimation, spatial relationships, object localization, spatial counting, etc. It is not directly for real robotics, it is relevant to embodied AI because spatial reasoning is important for embodied agents, so I can understand the relavance, but still not a good justification.

**Reviewer Scores:**

The initial scores are 6 4 4 6, but there are no discussions started unfortunately. The authors actively joined the discussions, and the mertis of their rebuttals are summarized above.

---

### Decision · Program_Chairs · 2026-01-26

Reject